# Global hydro-climatic biomes identified via multi-task learning

Christina Papagiannopoulou[1], Diego G. Miralles[2], Matthias Demuzere[2], Niko E. C. Verhoest[2], and Willem Waegeman[1]

[1]Department of Data Analysis and Mathematical Modelling, Ghent University, Belgium,
[2]Laboratory of Hydrology and Water Management, Ghent University, Belgium

**Correspondence:** Christina Papagiannopoulou (christina.papagiannopoulou@ugent.be)

**Abstract.** The most widely-used global land cover and climate classifications are based on vegetation characteristics and/or climatic conditions derived from observational data. However, these classification schemes do not directly stem from the characteristic interaction between the local climate and the biotic environment. In this work, we model the dynamic interplay between vegetation and local climate in order to delineate ecoregions that share a coherent response to hydro-climate variability. Our novel framework is based on a multi-task learning approach that discovers the spatial relationships among different locations by learning a low-dimensional representation of predictive structures. This low-dimensional representation is combined with a clustering algorithm that yields a classification of biomes with coherent behaviour. Experimental results using global observation-based data sets indicate that, without the need to prescribe any land cover information, the identified regions of coherent climate-vegetation interactions agree well with the expectations derived from traditional global land cover maps. The resulting global 'hydro-climatic biomes' can be used to analyse the anomalous behaviour of specific ecosystems in response to climate extremes and to benchmark climate-vegetation interactions in Earth system models.

## 1 Introduction

Approaches which aim to define regions with similar biophysical characteristics are commonly known as land cover classification schemes, and are widely used in multiple geoscientific disciplines. Land cover classifications are crucial to enable a better understanding of the spatial variability of the land surface, which can be a first and necessary step towards understanding complex spatio-temporal interactions among different environmental variables (Feddema et al., 2005). Traditional land use/land cover (change) classifications are typically based on spectral information from the land-surface coming from satellites (Loveland and Belward, 1997; Congalton et al., 2014). Amongst the most well-known and widely used are the International Geosphere-Biosphere Program DISCover Global 1km Land Cover classification (IGBP-DIS) (Loveland et al., 2000), Global Land Cover 2000 (Bartholomé and Belward, 2005) and more recently the land cover map developed within the European Space Agency's Climate Change Initiative (ESA CCI) (Poulter et al., 2015; Li et al., 2018). Similarly, climate classification schemes cluster regions with similar climate conditions and are also widely used to stratify geographical regions with different climatic expectations (Baker et al., 2009; Brugger and Rubel, 2013; Garcia et al., 2014; Herrando-Pérez et al., 2014). Here, the best known is probably the Köppen-Geiger climate classification (Köppen, 1936), which has been modified many times in recent decades (e.g. Thornthwaite, 1943; Trewartha and Horn, 1980; Feddema, 2005; Kottek et al., 2006; Peel et al., 2007). Yet to

date, dynamics in these climate regimes are used as diagnostic of climate change by exploring their shifting boundaries (e.g. Diaz and Eischeid, 2007; Chen and Chen, 2013; Zhang and Yan, 2014a, b; Spinoni et al., 2015; Chan and Wu, 2015) or as a means to predict future climatic zone distributions using climate projections (e.g. Hanf et al., 2012; Gallardo et al., 2013; Mahlstein et al., 2013).

In recent years, the exponential advance in Earth observation research has made climate science one of the most data-rich scientific domains (Faghmous and Kumar, 2014). As such, data-driven methods have become popular in their use for land cover and climate classifications. For instance, Lund and Li (2009) proposed a new distance measure to define seasonal means and autocorrelations of climatic time series from weather stations, and grouped the stations using a hierarchical agglomerative

clustering. Zscheischler et al. (2012) also stressed the importance of unsupervised methods for tasks such as the classification of the land surface into zones with different climate and vegetation characteristics. Metzger et al. (2012) applied an alternative data-driven approach on climate and vegetation data that used principal component analysis (PCA) to discover informative structures in the data. In this method, the principal components of the initial climate–vegetation data set were applied as input to a clustering algorithm. Interesting results in the same direction can be attributed to Netzel and Stepinski (2016, 2017),

who used distance measures of climatic variables, such as dynamic time warping, coming from the time series analysis in a data mining approach. In addition, temporal change in climate zones has been explored in the same context via clustering algorithms, such as k-means (Zhang and Yan, 2014a, b). Finally, data-driven methods have been also applied for the biome classification task, which has been commonly treated as an object recognition problem using remote sensing data. In this case, techniques coming from computer vision are frequently applied (Mekhalfi et al., 2015; Chen and Tian, 2015). Following the

progress in computer science, neural networks and deep learning approaches are also becoming popular for this kind of tasks in recent years, making the whole procedure even more automated (Scott et al., 2017; Xu et al., 2018).

Previous studies rely on spectral information, supervised techniques or clustering approaches, which are applied to observations of climate variables and/or vegetation characteristics. However, these classification schemes are not based on the type of

response of vegetation to climate dynamics. Recent advances in understanding vegetation response to climate variability highlight the importance of revealing the sensitivity of ecosystems to climate conditions, see Nemani et al. (2003); De Keersmaecker et al. (2015); Seddon et al. (2016); Papagiannopoulou et al. (2017b); Liu et al. (2018). Therefore, a step beyond these previous studies is a spatial characterization of the vegetation dynamics that are induced by climate variability, so that ecosystems of similar response to climate anomalies can be unveiled. This objective could be tackled by geostatistical approaches, such as

geographically weighted regression (GWR) (Brunsdon et al., 1996), which assume that neighboring pixels have a similar behaviour with respect to specific variables; these methods have already been applied in studies with a regional focus (Propastin et al., 2008; Zhao et al., 2015; Georganos et al., 2017). However, here, we aim to avoid neighborhood assumptions and focus on the discovery of relationships between pixels based on the similarity in their modelled climate–vegetation interaction, acknowledging that global ecosystems may experience similar interactions even if they are remotely located from each other. A

previous effort towards detecting regions with similar vegetation response to climate involves the work of Ivits et al. (2014),

where PCA is performed on the data matrix of drought anomalies and vegetation state, and a clustering is applied to the correlation coefficients based on the spatio-temporal patterns obtained by PCA. However, in this study, the interaction between climate and vegetation is not explicitly learned, nor the causes behind vegetation changes are inferred in a predictor–target framework.

Here, we introduce for the first time (to the best of our knowledge) a data-driven approach that aims to quantify the response of vegetation to local climate variables in a supervised setting at a global scale, and use this information to define ecoregions of consistent behaviour against hydro-climatic variability. In simple terms, our framework results in regions where vegetation responds similarly to the dynamics in temperature, soil moisture, incoming radiation, etc. The proposed framework relies on predictive modelling and clustering techniques and builds further upon recent work in which we investigated the global response of vegetation to local climate by applying machine learning algorithms in a Granger causality setting (Papagiannopoulou et al., 2017a, b). Since here we aim to exploit the relationships between different pixels – instead of modelling each pixel separately as in our previous work – we propose the use of multi-task learning (MTL) methods (Caruana, 1997). These methods are commonly used for solving multiple related tasks: considering as one task the prediction of vegetation in one location and as multiple tasks the prediction of vegetation in multiple locations, we can model our problem by using an MTL approach. First, we apply an MTL approach which tries to unveil low-dimensional common predictive structures and exploit the relationships among them. Second, we employ a clustering technique on these informative structures, which is applied on a lower-dimensional space (Sect. 2). This clustering technique is known as spectral clustering (Ng et al., 2002), and is one of the core assets of our framework. We refer to the emergent regions of coherent vegetation–climate behaviour as *hydro-climatic biomes* (Sect. 3).

## 2 Methodology

### 2.1 Data sets

We have built a large database of global climate and vegetation data that will be used in the context of our framework. These data are described in detail in Papagiannopoulou et al. (2017a) and are mostly based on satellite and/or *in situ* observations. The database spans a 30-year period (1981-2010) at monthly temporal resolution and 1° latitude-longitude spatial resolution. The most important climatic and environmental drivers of vegetation are included, namely: (i) land surface temperature, (ii) near-surface air temperature, (iii) longwave/shortwave surface radiative fluxes, (iv) precipitation, (v) snow water equivalent, and (vi) soil moisture. To characterise vegetation, we use the Global Inventory Modelling and Mapping Studies (GIMMS) NDVI 3g data set (Tucker et al., 2005). The target variable of our machine-learning framework is the de-trended seasonal NDVI anomalies. These are calculated through a simple linear de-trending and a multi-year average for each month of the year to capture the seasonal expectation – see Papagiannopoulou et al. (2017a) for more details. All other data sets, describing the multi-month local climate variability over the three-decade period, are used as predictor variables.

In addition, a wide range of 'high-level features' have been hand-crafted from the raw time series of predictors, and used as well as predictor variables. As such, our set of predictive features includes not just the raw data time series of each cli-

mate/environmental variable, but also: seasonal anomalies, de-trended seasonal anomalies, lagged variables, past cumulative variables, and extreme indices – see Papagiannopoulou et al. (2017a). The cumulative variables capture the climatic conditions up to present time; an example would be the precipitation of the last (e.g.) three months. Extreme indices include maximum/minimum values, consecutive dry days, values for specific percentiles, etc. The use of these non-linear features (non-linear due to the way that have been calculated) greatly improves causal inference and helps characterise non-linear relationships between climate and vegetation dynamics, as shown in our recent work (Papagiannopoulou et al., 2017a). For further discussion about the importance of this higher-level feature representation adopted in our framework, we refer the reader to Sect. S1 of the Supplementary material.

## 2.2  Pixel-based approach: single-task learning

In our study, we use information on climate and vegetation variables at specific time points and locations. Formally, we consider a spatio-temporal data set $D = \{(\mathbf{X}^{(1)}, \boldsymbol{y}^{(1)}), (\mathbf{X}^{(2)}, \boldsymbol{y}^{(2)}), ..., (\mathbf{X}^{(L)}, \boldsymbol{y}^{(L)})\}$, with $L$ being the number of different locations and $(\mathbf{X}^{(l)}, \boldsymbol{y}^{(l)})$ the tuple of the predictor variables and the target variable of each location $l$. We denote $D^{(l)} = \{(\boldsymbol{x}_i^{(l)}, y_i^{(l)})\}_{i=1,...,N}$ the observations of a location $l$ while the input feature vectors (i.e. the set of climatic variables) are denoted as a matrix $\mathbf{X}^{(l)} = [\boldsymbol{x}_1^{(l)}, ..., \boldsymbol{x}_N^{(l)}]^T$ and the corresponding target values as $\boldsymbol{y}^{(l)} = [y_1^{(l)}, ..., y_N^{(l)}]^T$ (i.e., the NDVI anomalies). Specifically, $\mathbf{X}^{(l)} \in \mathbb{R}^{N \times d}$ is the matrix of the predictor variables with $d$ being the number of predictors, and $\boldsymbol{y}^{(l)} \in \mathbb{R}^N$ the response time series (i.e., NDVI seasonal de-trended anomalies), where $N$ denotes the number of discrete time stamps, i.e., the length of the time series. In this setting, a straightforward approach is to tackle each regression problem in each location $l$ separately, i.e., by independently training one model for each location (Papagiannopoulou et al., 2017a). That way, for every pixel only the data of that particular location $l$ is used ($(\mathbf{X}^{(l)}, \boldsymbol{y}^{(l)}), l = 1, ..., L$), not attempting to utilize the data from other regions where the target variable might have a similar response to the predictors.

We can start by defining regions of similar climate–vegetation dynamics with the most naive approach: the relationship between climate and vegetation can be caught by the weights of a simple regression model, i.e., the regression coefficients of the predictor variables. Specifically, if one defines a simple linear regression model for a location $l$, the model for the $l^{\text{th}}$ location is given by $f^{(l)}(\boldsymbol{x}_i^{(l)}) = \boldsymbol{w}^{(l)} \boldsymbol{x}_i^{(l)}$, with $\boldsymbol{x}_i^{(l)}$ being the input data (i.e., one observation) and $\boldsymbol{w}^{(l)}$ being the weight vector learned for particular location $l$, which describes the importance of each input variable for the target – see Fig. 1a. Even though one can assume that these weight vectors can be similar for regions in which the response of vegetation to climate is similar, the information from these other regions is not used in the prediction (i.e. each regression is applied at each individual pixel separately). This is despite the fact that these locations could be subsequently grouped (e.g., based on a similarity measure of their weight vectors) into wider regions that one may assume that share common climate–vegetation dynamics. Note also that the information captured by each weight vector $\boldsymbol{w}^{(l)}$ should be sufficient which means that it is necessary for the models to have a good generalization performance.

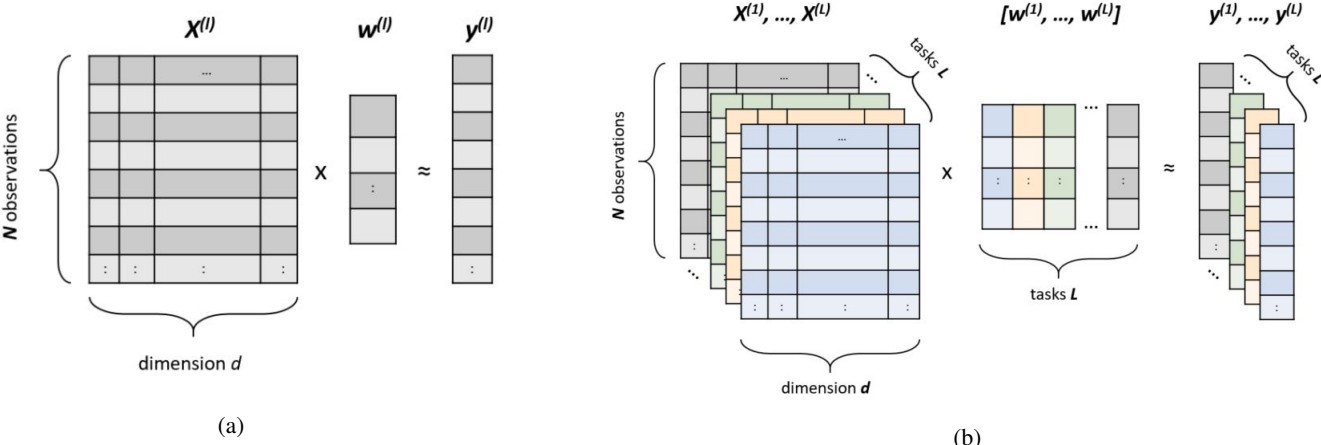

(a)                                                                      (b)

**Figure 1.** Graphical representation of two learning approaches. (a) A single-task learning approach in which each pixel is treated separately. For each pixel $l$ there is an input data set $\mathbf{X}^{(l)} \in \mathbb{R}^{N \times d}$, with $N$ being the number of observations and $d$ being the number of predictors, and a target vector $\boldsymbol{y}^{(l)} \in \mathbb{R}^N$. The vector $\boldsymbol{w}^{(l)} \in \mathbb{R}^d$ represents the weight vector learned by the model. (b) A multi-task learning approach in which the models of $L$ tasks are simultaneously learned. The input of the method is the data sets $\mathbf{X}^{(1)}, \mathbf{X}^{(2)}, ..., \mathbf{X}^{(L)}$ of all locations (i.e., all global land pixels). The corresponding target vectors are denoted with $\boldsymbol{y}^{(1)}, \boldsymbol{y}^{(2)}, ..., \boldsymbol{y}^{(L)}$. The weight matrix $[\boldsymbol{w}^{(1)}, \boldsymbol{w}^{(2)}, ..., \boldsymbol{w}^{(L)}] \in \mathbb{R}^{d \times L}$ contains the weight vectors for all tasks.

## 2.3  Exploiting spatial relationships: multi-task learning

Unlike the single-task learning models described above, that only take the data of each particular location into account, MTL models extract information of data sets with similar characteristics from other locations. As such, they can be expected to generalize better and give a higher predictive performance on unseen data. Specifically, by using the MTL approach, the generalization of the model improves if the dataset of each task is expanded by observations from highly related tasks. This is crucial, especially in cases where the number of training instances per task is limited. The basic idea that underlines the MTL modelling approach is the learning of a separate model for each task and not a unique model trained on a concatenated set of observations of all tasks. Note that in our spatio-temporal data sets, each location can be seen as a different task, and that neighbouring (or distant) locations with similar climate–vegetation interactions will tend to have similar (yet not identical) behaviour. In light of this observation, MTL seems to be a quite natural modelling approach to explore the interaction between climate and vegetation in different locations.

The idea of MTL is not new (Baxter, 1997; Caruana, 1997; Baxter et al., 2000), and it has been applied in many machine-learning applications in medical sciences (Bi et al., 2008; Zhang et al., 2012) and computer vision (Zhang et al., 2014). It has also been used in climate science to improve the way multiple Earth System Models (ESMs) outputs are combined, by treating the locations as different tasks (Subbian and Banerjee, 2013; McQuade and Monteleoni, 2013). In these studies, the idea is that in neighbouring locations (pixels which are close to each other), similar ESMs tend to have similar performance. A recent study proposed a hierarchy of tasks, in which at a first level, tasks of each location are trained into an MTL setting, while at a second

level, tasks of each variable are sharing information (Gonçalves et al., 2017). In addition, for modelling spatio-temporal data, Xu et al. (2016) introduced an MTL framework in which local models share a common representation based on the spatial autocorrelation. Although this kind of modelling is becoming more common in climate science (i.e., Subbian and Banerjee (2013); McQuade and Monteleoni (2013); Gonçalves et al. (2017); Xu et al. (2016)), it has not been combined (to the best of our knowledge) with clustering approaches in the context of mapping land cover nor climate–vegetation dynamics.

In this work, we focus on MTL methods that can discover the relationship between different tasks (locations) and recover strong predictive structures of the vegetation response to climate. These are then used to conform hydro-climatic biomes, i.e., regions of coherent vegetation behaviour with respect to climate variability (see Sect. 3.3). To this end, we use the same notation as before by denoting $\mathbf{X}^{(l)} \in \mathbb{R}^{N \times d}$ as input data matrix of the predictor variables, $\boldsymbol{y}^{(l)} \in \mathbb{R}^N$ as the target vector for each location $l$ and $\boldsymbol{w}^{(l)} \in \mathbb{R}^d$ in which each value corresponds to a weight. We define as $[\boldsymbol{w}^{(1)}, \boldsymbol{w}^{(2)}, ..., \boldsymbol{w}^{(L)}] \in \mathbb{R}^{d \times L}$ the weight matrix of all locations such that the $\boldsymbol{w}^{(l)}$ vector is the $l^{\text{th}}$ column of the $[\boldsymbol{w}^{(1)}, \boldsymbol{w}^{(2)}, ..., \boldsymbol{w}^{(L)}]$ matrix – see a graphical representation of the notation in Fig. 1b. Given a loss function $\mathcal{L}$ (e.g., the squared error loss), the multi-task minimization problem is formulated as:

$$\min_{w^{(1)}, ..., w^{(L)}} \sum_{l=1}^{L} \sum_{i=1}^{N} \mathcal{L}(\boldsymbol{w}^{(l)} \boldsymbol{x}_i^{(l)}, y_i^{(l)}) + \Omega(\boldsymbol{w}^{(1)}, ..., \boldsymbol{w}^{(L)}) \tag{1}$$

where $\Omega(\boldsymbol{w}^{(1)}, ..., \boldsymbol{w}^{(L)})$ is a factor which controls the relatedness among the tasks. In our setting, we assume that there is no prior knowledge about the relationship of the tasks (locations) and we aim to apply a method that can discover these relationships.

In literature, there are many MTL methods that are trying to do two things simultaneously: learn a weight matrix $[\boldsymbol{w}^{(1)}, \boldsymbol{w}^{(2)}, ..., \boldsymbol{w}^{(L)}]$ and another matrix which captures the task relationships simultaneously (Ando and Zhang, 2005; Chen et al., 2009; Zhou et al., 2011). In real applications, there are scenarios where the tasks of an MTL problem follow a specific structure, i.e., some tasks are more related whereas some others are unrelated. In order to identify this group structure, researchers have developed various methods which have been referred to as clustered multi-task learning (CMTL) methods (Zhou et al., 2011). For instance, Xue et al. (2007) proposed a method which uses a Dirichlet process-based statistical model to identify similarities between related tasks, while Jacob et al. (2009) introduced a framework which identifies groups of tasks and performs the learning at once. In the same direction, Wang et al. (2009) used an inter-task regularization term to take into consideration tasks which have been grouped in the same cluster in a semi-supervised setting. More recently, Barzilai and Crammer (2015) suggested a method which assigns explicitly each task to a specific cluster, building a single model for each task by using linear classifiers which are combinations of some basis. An alternative approach has been proposed by Zhou et al. (2011) in which the structure of the task relatedness is unknown and is learned during the training phase. Interestingly, when case-specific conditions are fulfilled, this method is equivalent to the method by Ando and Zhang (2005), known as the Alternative Structure Optimization (ASO), which belongs to the category of MTL methods that assume the existence of a shared low-dimensional representation among the tasks. The name of the method indicates that an alternating optimization procedure is involved during the learning process since the weight matrix and the matrix which captures the shared low-dimensional

representation are learned simultaneously. Typically, in these procedures, the optimization of each part is separately performed while the other part remains fixed. In our work, we apply the ASO method due to its simplicity and the fact that it does not need a lot of iterations to capture the information about the task relatedness that is needed. This is crucial for our application, since the large size of the global database we use (Papagiannopoulou et al., 2017a) puts severe limitations to the choice of method. Another aspect is that by learning this low-dimensional representation we can have a visual inspection of the "most predictive common structures" for each region. In the following section we explain in detail the ASO method used in our setting.

## 2.4 Learning predictive structures from multiple tasks

The ASO algorithm proposed by Ando and Zhang (2005) learns common predictive structures from multiple related tasks that are assumed to share a low-dimensional feature space. Specifically, by applying this method, one learns one model function for each individual task and the learned weight vector is decomposed into two parts: (a) a high-dimensional space, and (b) a shared low-dimensional space based on a feature map learned during the process. This feature map is a matrix which serves as a link between a high-dimensional space and a low-dimensional space. In our case, $L$ predictor functions $\{f^{(l)}\}_{l=1}^{L}$ are simultaneously learned by exploiting the shared feature space that underlines all tasks. This low-dimensional feature space is expressed in a simple linear form of a low-dimensional feature map $\boldsymbol{\Theta}$ across the $L$ tasks. Mathematically, the function $f^{(l)}$ can be written as:

$$f^{(l)}(\boldsymbol{x}) = \boldsymbol{w}^{(l)}\boldsymbol{x}_i^{(l)} = \boldsymbol{u}^{(l)}\boldsymbol{x}_i^{(l)} + \boldsymbol{v}^{(l)}\boldsymbol{\Theta}\boldsymbol{x}_i^{(l)} \tag{2}$$

with $\boldsymbol{\Theta} \in \mathbb{R}^{h \times d}$ being a parameter matrix with orthonormal row vectors, i.e., $\boldsymbol{\Theta}\boldsymbol{\Theta}^T = \mathbf{I}$, where $h$ is the dimensionality of the shared feature space, and $\boldsymbol{w}^{(l)}, \boldsymbol{u}^{(l)}$ and $\boldsymbol{v}^{(l)}$ are the weight vectors for the full feature space, the high-dimensional one (initial dimension $d$), and the shared low-dimensional one (based on the $h$ parameter), respectively. As mentioned before, the ASO method is equivalent to the CMTL method (Zhou et al., 2011), under a specific condition: that the parameter $k$, which symbolizes the number of clusters in the CMTL approach, is equal to the parameter $h$ of the ASO method. This condition determines the number of clusters that should be used in the clustering phase of our framework, because the objective of ASO is optimized based on the value of the parameter $h$. We reconsider this equivalence in Sect. 3.2 where we discuss about the number of clusters that should be identified based on our analysis.

Formally, ASO can be formulated as the following optimization problem:

$$\min_{\{\boldsymbol{w}^{(l)}, \boldsymbol{v}^{(1)}\}, \boldsymbol{\Theta}\boldsymbol{\Theta}^T = \mathbf{I}} \sum_{l=1}^{L} \left( \sum_{i=1}^{N} \mathcal{L}(\boldsymbol{w}^{(l)}\boldsymbol{x}_i^{(l)}, y_i^{(l)}) + \lambda^{(l)} \left\| \boldsymbol{u}^{(l)} \right\|_2^2 \right) \tag{3}$$

with $\left\| \boldsymbol{u}^{(l)} \right\|_2^2$ being the regularization term ($\boldsymbol{u}^{(l)} = \boldsymbol{w}^{(l)} - \boldsymbol{\Theta}^T \boldsymbol{v}^{(l)}$) that controls the task relatedness among $L$ tasks, $(\boldsymbol{x}_i^{(l)}, y_i^{(l)})$ being the input vector and the corresponding target value of the $i^{\text{th}}$ observation in a particular location $l$, and $\lambda^{(l)}$ being a pre-defined parameter – see Fig. 2 for the graphical representation of the notation. During the learning process the weight matrix $[\boldsymbol{w}^{(1)}, \boldsymbol{w}^{(2)}, ..., \boldsymbol{w}^{(L)}]$ and the matrix $\boldsymbol{\Theta}$, which captures the shared low-dimensional representation, are learned simultaneously. The regularization term $\left\| \boldsymbol{u}^{(l)} \right\|_2^2$, based on the value of the parameter $\lambda$, penalizes the differences between the weights on the initial high-dimensional space and the weights on the low-dimensional space parameterized by $\boldsymbol{\Theta}$.

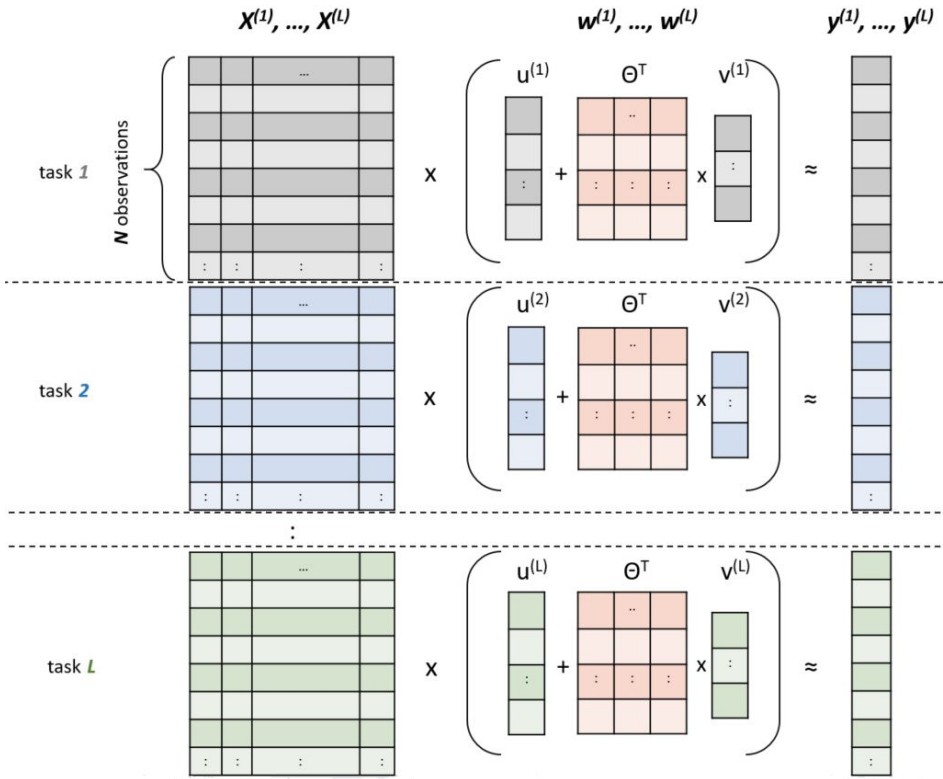

**Figure 2.** Graphical representation of the ASO method. The input of the method is the data sets $\mathbf{X}^{(1)}, \mathbf{X}^{(2)}, ..., \mathbf{X}^{(L)}$ of all locations. The corresponding target vectors are denoted with $\boldsymbol{y}^{(1)}, \boldsymbol{y}^{(2)}, ..., \boldsymbol{y}^{(L)}$. The weight vector $\boldsymbol{w}^{(l)} \in \mathbb{R}^d$ of the full space is decomposed in two parts; to the weight vector $\boldsymbol{u}^{(l)} \in \mathbb{R}^d$ of the high-dimensional space and the weight vector $\boldsymbol{v}^{(l)} \in \mathbb{R}^h$ of the low-dimensional one. The low-dimensional feature map $\boldsymbol{\Theta}^T \in \mathbb{R}^{d \times h}$ is common for all the tasks.

There are several ways of solving the optimization problem in Eq. (3) (Ando and Zhang, 2005). Our main purpose is to extract the shared feature space $\boldsymbol{\Theta}$ in order to apply a clustering on the low-dimensional feature space. In this feature space, locations with similar predictive structures will be grouped into the same broader region. For this reason, we adopt the Singular Value Decomposition (SVD)-based ASO algorithm, proposed by Ando and Zhang (2005), which achieves good performance

5    even on the first iteration of the method. As mentioned before, this is crucial to our application given the large number of tasks and the high-dimensional data sets. The steps of the SVD-based ASO are presented in Algorithm 1.

The SVD-based ASO method can be interpreted as a dimensionality reduction technique applied to the model space (i.e., weights). It should be stressed here that this method must not be confused with PCA, which is usually employed on the data space (input space of predictors) (Metzger et al., 2012; Ivits et al., 2014). The goal of the ASO method is to detect the principal

10   components of the parameter matrix, while PCA identifies the principal components of the input data $\mathbf{X}$. The goal of the ASO method can be achieved by considering the models of multiple tasks as samples of their own distribution. Therefore, these samples can only be formed by using an MTL approach, in which there is access to the models from multiple learning tasks.

**Algorithm 1** SVD-ASO

Input: training data $D^{(l)} = \{(\boldsymbol{x}_i^{(l)}, y_i^{(l)})\}_{i=1,...,N}$, where $l = 1,...,L$

Parameters: $h$ and $\boldsymbol{\lambda} = \{\lambda^{(1)},...,\lambda^{(L)}\}$

Output: $\boldsymbol{\Theta} \in \mathbb{R}^{h \times d}$ and $\mathbf{V} = [\boldsymbol{v}^{(1)},...,\boldsymbol{v}^{(L)}]^T \in \mathbb{R}^{L \times h}$

Initialize: $\boldsymbol{w}^{(l)} = 0, l = 1,...,L$, and $\boldsymbol{\Theta}$ to random

**repeat**

    **for** $l = 1$ **to** $L$ **do**

        with fixed $\boldsymbol{\Theta}$ and $\boldsymbol{v}^{(l)} = \boldsymbol{\Theta}\boldsymbol{w}^{(l)}$, solve the optimization problem of Eq. (3) for $\boldsymbol{u}^{(l)}$:

        $\text{argmin}_{\boldsymbol{u}^{(l)}} \sum_{i=1}^{N} \mathcal{L}(\boldsymbol{u}^{(l)}\boldsymbol{x}_i^{(l)} + (\boldsymbol{v}^{(l)}\boldsymbol{\Theta})\boldsymbol{x}_i^{(l)}, y_i^{(l)}) + \lambda^{(l)} \left\| \boldsymbol{u}^{(l)} \right\|_2^2$

        $\boldsymbol{w}^{(l)} = \boldsymbol{u}^{(l)} + \boldsymbol{\Theta}^T\boldsymbol{v}^{(l)}$

    **end for**

    Apply an SVD decomposition on $\mathbf{W} = [\sqrt{\lambda^{(1)}}\boldsymbol{w}^{(1)},...,\sqrt{\lambda^{(L)}}\boldsymbol{w}^{(L)}]$:

    $\mathbf{W} = \mathbf{V_1}\mathbf{D}\mathbf{V_2}^T$ (with diagonals of $\mathbf{D}$ in descending order)

    $\boldsymbol{\Theta} = \mathbf{V_1}^T[:h,:]$ // update $\boldsymbol{\Theta}$ to the first $h$ rows of $\mathbf{V_1}^T$

**until** convergence

Moreover, in our work, we explicitly consider the climatic variables as predictors and the vegetation variable as target variable, and we learn the relationship between them in a supervised setting. As such, the regions that we define rely on the relationship between climate and vegetation in a prediction setting, and the clustering is calculated based on similarity of this relationship (i.e. the model coefficients for different locations), see Sect. 2.5 for more details. As such, we learn relationships between climate and vegetation in a supervised setting, whereas PCA-based methods (Metzger et al., 2012; Ivits et al., 2014) are fully unsupervised. In our study the SVD decomposition is used as part of the optimization algorithm, thus in a supervised setting. In this setting, the model weights are optimized based on a given training set. Therefore, the discovered structures are obtained during the training process.

To clarify the notation used in the ASO method, we intuitively explain the symbolization of the method in relation to our specific setting: the problem of detecting locations with similar climate–vegetation dynamics. As mentioned above (Sect. 2.2 and 2.3), the input features that constitute the $\mathbf{X}^{(l)} \in \mathbb{R}^{N \times d}$ matrix consist of the climatic predictor variables, i.e., the extreme indices, lagged variables, etc., calculated based on raw climatic time series of a certain location $l$. The dimensions $N$ and $d$ correspond to the number of observations, i.e., the length of the time series and the number of predictor variables, respectively. The target variable for a particular location $l$, which is the NDVI anomalies, is symbolized with $\boldsymbol{y}^{(l)} \in \mathbb{R}^N$. As such, an observation of a certain location $l$ at a particular timestamp $i$ is denoted as a pair $(\boldsymbol{x}_i^{(l)}, y_i^{(l)})$. The goal of the ASO method is to learn the weight matrix $[\boldsymbol{w}^{(1)}, \boldsymbol{w}^{(2)},...,\boldsymbol{w}^{(L)}]$, i.e., a single weight vector $\boldsymbol{w}^{(l)}$ for each location $l$. This weight vector $\boldsymbol{w}^{(l)}$ is able to capture the relationship between the predictor variables and the target, i.e., the climatic variables and the NDVI anomalies. Therefore, climatic predictors that are more important for vegetation anomalies correspond to higher absolute values in the weight vector $\boldsymbol{w}^{(l)}$. As a result, locations with similar weights are considered as regions where vegetation responds to

climate in a similar way. As described in a previous paragraph of this section, the ASO method assumes that the weight vectors $\boldsymbol{w}^{(l)}$ consist of two parts the $\boldsymbol{u}^{(l)}$ and the $\boldsymbol{v}^{(l)}\boldsymbol{\Theta}$. These two parts are learned simultaneously in Algorithm 1 in an alternating fashion. The first part, i.e., the $\boldsymbol{u}^{(l)} \in \mathbb{R}^d$ belongs to the high-dimensional space, the initial one, which is equal to $d$. This part expresses the location-specific part of the weight vector, i.e., the deviation of each location's weight vector from the weights learned in a lower dimensional space. The second part consists of the matrix $\boldsymbol{\Theta} \in \mathbb{R}^{h \times d}$ that represents the map from the initial dimension $d$ to the lower dimension $h$ and the weight vector $\boldsymbol{v}^{(l)} \in \mathbb{R}^h$. The map matrix $\boldsymbol{\Theta}$ is common for all the locations (tasks) and can be learned across them due to the MTL approach. The weight vector $\boldsymbol{v}^{(l)}$ represents the projection of the initial weights to a low-dimensional space $h$. Intuitively, this second part of the weight decomposition expresses the coarsest and most important part of weights, since it detects the most important structures through the map matrix $\boldsymbol{\Theta}$. The matrix $\mathbf{V} = [\boldsymbol{v}^{(1)}, ..., \boldsymbol{v}^{(L)}]^T \in \mathbb{R}^{L \times h}$ denotes the representation of the models in the low-dimensional space $h$ for the $L$ locations.

## 2.5 Land classification: clustering highly-predictive structures

Clustering in machine learning is the task of grouping a set of samples in such a way that those samples that belong to the same group (cluster) are more similar with respect to a specific criterion than to samples that belong to other groups. Clustering techniques are usually based on a distance (or similarity) measure, which is calculated among the samples and/or group of samples. There are several clustering approaches and an in-depth review can be found in Xu and Tian (2015).

It is known that in high-dimensional spaces, the distance measures are not able to capture well the differences between pairs of samples, thus clustering algorithms tend to perform better in lower dimensional spaces. In our setting, we learn the common feature map $\boldsymbol{\Theta} \in \mathbb{R}^{h \times d}$ and the $\mathbf{V} = [\boldsymbol{v}^{(1)}, ..., \boldsymbol{v}^{(L)}]^T \in \mathbb{R}^{L \times h}$ matrix, which is the representation of the models in this low-dimensional space, using the SVD-ASO method – see Sect. 2.4. The $\mathbf{V}$ matrix captures the information of the similar predictive structures among all the tasks, so similar tasks are closer in this low dimensional space and as a consequence, they have a similar representation (i.e. weights) in this matrix. That way, the clustering techniques based on distance calculations are applied on the more expressive low-dimensional space, resulting in a better performance. As it has been discussed in our previous work (Papagiannopoulou et al., 2017a), global climate-vegetation relationships are complex and non-linear. Here, if the $\mathbf{V}$ representation is expressive enough, the clustering method can group together locations with similar models, i.e., locations in which vegetation responds to climate in a similar non-linear way. Thus, it is first necessary to evaluate the quality of the learned matrix $\mathbf{V}$. The most straight forward way to do so, is by measuring the predictive performance of the MTL model in terms of e.g. $R^2$. If the predictive power of the model is strong, we can conclude that the $\mathbf{V}$ matrix is able to well-capture the relationships of each task with the highly predictive structures. So, given that the $\mathbf{V}$ representation is sufficiently learned from the data, we can apply any kind of clustering algorithm on the low-dimensional representation of matrix $\mathbf{V}$. This approach is also known as spectral clustering due to the fact that the clustering algorithm is applied on a reduced feature space, making the clustering results more robust.

In our application, we use a hierarchical agglomerative clustering approach (Ward, 1963) where the number of clusters is not predefined. In the hierarchical clustering approach, the result is usually depicted as a dendrogram in which the leaves represent the observations and the inner nodes correspond to the data clusters. The dendrogram branches are proportionally long to the

value of the intergroup dissimilarity. By defining this hierarchical form of the clustering result, one can define the number of clusters by cutting down vertically (or horizontally, depending on the view) the dendrogram in a point where the dissimilarity between the clusters is high and therefore the branches are longer – see Sect. 3.2 for the choice of the optimum number of clusters in our analysis.

## 2.6 Experimental setup

In all the experiments, we use as predictors all the climatic data sets and the features that we have constructed from them as well as the 12-lagged values of the target variable. A resulting number of 3,209 predictor (climate) variables is used, i.e., $d = 3,209$ in our setting. These variables constitute the input to our framework, i.e., the $\mathbf{X}^{(l)}, l = 1, ..., L$ data sets. As target variable, we use the NDVI seasonal anomalies calculated as in Papagiannopoulou et al. (2017a) and denoted as $\boldsymbol{y}^{(l)}, l = 1, ..., L$ for each location. For more details about the data sets in our setting see Sect. 2.1. We examine 13,072 land pixels where each pixel constitutes a single task in our MTL setting, i.e., $L = 13,072$. The dataset of each single task consists of 360 monthly observations given our 30-year study period, i.e., $N = 360$.

For the STL modelling, evaluated for comparison, we use the ridge regression for each location independently. Ridge regression is a linear model which uses an $\ell_2$ norm regularization term in order to shrink the weight coefficients towards zero and avoid over-fitting. In ridge regression the weight coefficients are fitted by solving the following optimization problem:

$$\min_{\boldsymbol{w}^{(l)}} \sum_{i=1}^{N} \mathcal{L}(\boldsymbol{w}^{(l)}\boldsymbol{x}_i^{(l)}, y_i^{(l)}) + \lambda ||\boldsymbol{w}^{(l)}||^2 \tag{4}$$

with $\lambda$ being a regularization parameter tuned using a separate validation set and $||\boldsymbol{w}^{(l)}||^2$ being a penalty term, i.e., the squared $\ell_2$ norm of the weight vector. Note that by splitting the original data set in three parts – (1) training set, (2) validation set, and (3) test set – we tune the parameters in a set of observations (validation set) that are not included in the final test set and achieve a fair evaluation of the model performance. The optimization problems of the SVD-ASO algorithm are solved by using the Limited-memory Broyden-Fletcher-Goldfarb-Shanno (L-BFGS) optimization algorithm.

## 3 Results and discussion

### 3.1 Single- versus multi-task learning model

In a first experiment, we compare the predictive performance of the STL model versus the MTL model. For the STL modelling, the ridge regression is used. For the MTL modelling, we apply the ASO-MTL model (Ando and Zhang, 2005) described in Sect. 2. We use a separate validation set to tune the regularization parameter $\lambda$ for both approaches. For the STL approach, we tune the $\lambda$ parameter for each location (task) separately, while for the MTL approach we use the same $\lambda$ value for all the tasks, taking into account the average performance across these tasks. For the ASO-MTL method, we have also experimented with the value of the $h$ parameter, which is the dimensionality of the shared feature space – see Sect. 3.2 for more details about the influence of this parameter on the clustering results. Finally, we evaluate the performance of both approaches in terms of $R^2$,

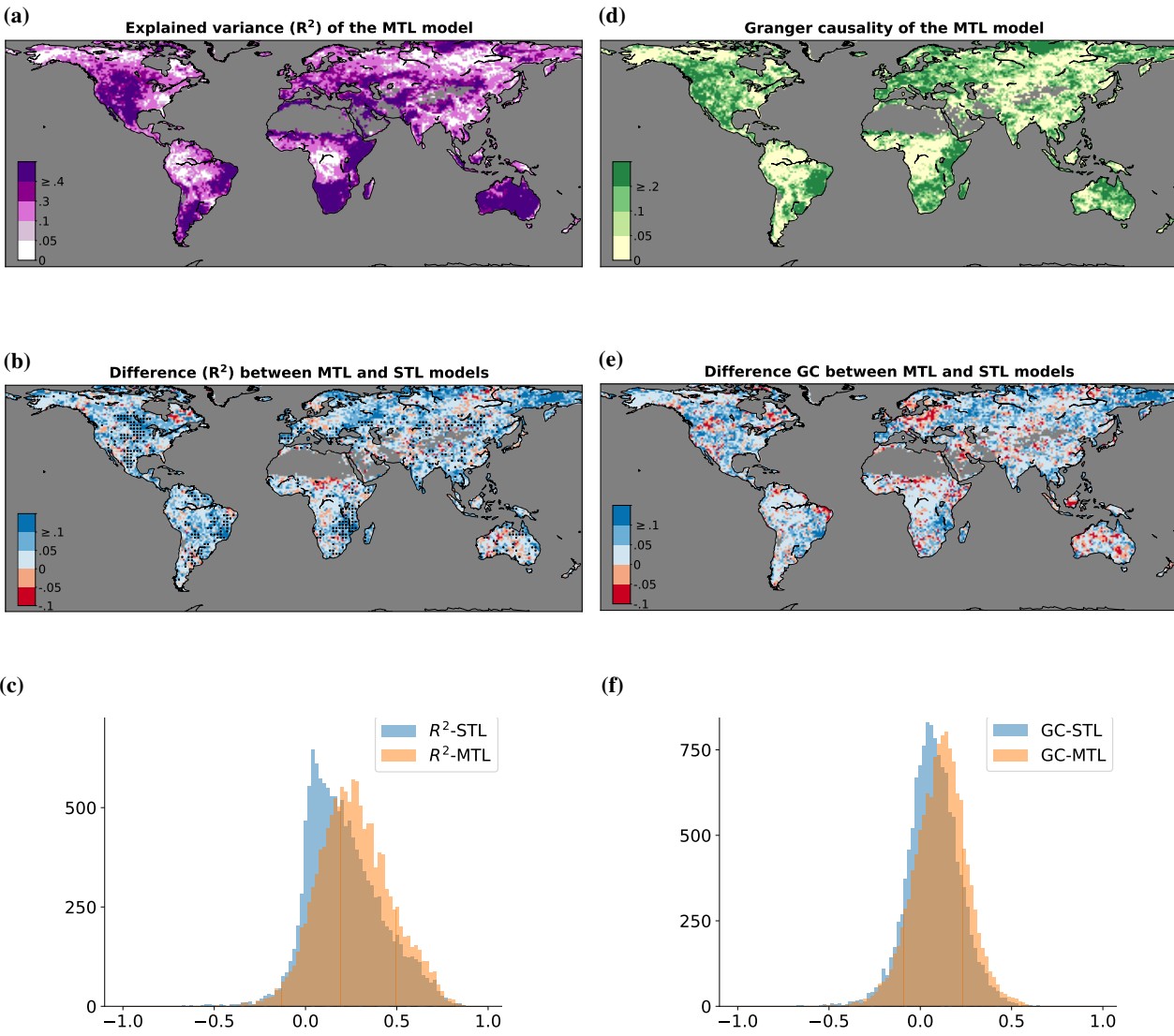

**Figure 3.** Comparison of the predictive performance between the STL and the MTL approaches. (a) Explained variance ($R^2$) of the NDVI monthly anomalies based on the MTL approach. (b) Difference in terms of $R^2$ between the MTL and the STL approaches; blue regions indicate a higher performance by the MTL. The dotted regions correspond to areas where the MTL model significantly outperforms the STL models based on the Diebold-Mariano statistical test (Diebold, 2015). (c) Comparison of the distributions of the $R^2$ scores in the STL and in the MTL setting; the blue histogram corresponds to the STL, and the orange one to the MTL approach. (d) Quantification of Granger causality for the MTL approach, i.e. improvement in terms of $R^2$ by the full MTL model with respect to the $R^2$ of the baseline MTL model that uses only past values of NDVI anomalies as predictors; positive values indicate Granger causality (Papagiannopoulou et al., 2017a). (e) Difference in terms of Granger causality between the MTL and the STL approaches; blue regions indicate a higher performance by the MTL. (f) Comparison of the distributions of the Granger causality in the STL and in the MTL setting; the blue histogram corresponds to the STL, and the orange one to the MTL approach.

as in Papagiannopoulou et al. (2017a). Figure 3 depicts the result of our comparison. Figure 3a shows the $R^2$ of the ASO-MTL model while Fig. 3b highlights the difference in predictive performance of the MTL model in comparison with the STL model. As shown in Fig. 3b, in almost all regions of the world, the predictive performance increases substantially compared to the STL approach. In fact, over extensive regions (40% of the study area), more than 5% of the variability in NDVI is explained by the spatial structure of the data. In statistical terms, this implies the existence of a hidden structure between the different locations (tasks), which is informative with respect to our target variable. The dotted regions in Fig. 3b correspond to areas where the MTL model significantly outperforms the STL models based on the Diebold-Mariano statistical test, which compares model predictions (Diebold, 2015). For the statistical test, we use the False Discovery Rate (Benjamini and Hochberg, 1995) method to correct the p-values at level 0.05 due to the multiple-hypothesis testing setting.

Additionally, Fig. 3a shows that more than 40% of the mean monthly vegetation dynamics can be explained by climate variability in some regions. In particular, in regions such as Australia, Africa and Central and North America, the predictive power of the model is stronger in terms of $R^2$, following the same pattern and scoring similar $R^2$ values as the random forest approach by Papagiannopoulou et al. (2017a). To deepen on the performance difference between the two approaches, the $R^2$ scores are presented as two different distributions in Fig. 3c. The blue histogram corresponds to the distribution of the $R^2$ scores of the STL approach, while the orange one corresponds to the distribution of the $R^2$ scores of the MTL approach. As it can be observed, the distribution of the $R^2$ scores is shifted to the right for the MTL, meaning that values are typically greater than those derived from the STL approach. Moreover, the skew towards the left in the blue histogram, with values close to zero, is an indication of the near-zero performance of the STL models in many locations. The Wilcoxon paired statistical test (Demšar, 2006) confirms that the results of the two approaches are overall statistically different (p-value $< 10^{-9}$).

Since we are ultimately interested in investigating regions of coherent impact of climate variability on vegetation dynamics, we also evaluate the ability of the MTL model to detect Granger-causal effects of climate on vegetation. For a detailed description of the Granger causality modelling framework we direct the reader to Papagiannopoulou et al. (2017a). This point is crucial to understand the extent to which the climatic predictors carry additional information about the dynamics in vegetation that is not contained in the past vegetation signal itself. The results of applying the Granger causality analysis using MTL modelling are shown in Figure 3d, which illustrates results of the full MTL model compared to the baseline MTL model. This baseline model only uses previous values of NDVI to predict monthly NDVI anomalies (Papagiannopoulou et al., 2017a). In this figure it becomes clear that climate dynamics Granger-cause monthly vegetation anomalies in most regions of the world, and the ability of the MTL model to detect deterministic relationships is evidenced. This is also confirmed by the Wilcoxon paired statistical test (p-value $< 10^{-9}$). On the other hand, the ability of the STL model to detect Granger-causal relationships is rather limited compared to that of the MTL model. Figure 3e depicts the result of the comparison, where in almost all regions the quantification of Granger causality of the MTL approach increases substantially compared to the one of the STL approach. Analogous to Fig. 3c, Fig. 3f compares the distributions of Granger causality (i.e., the difference in predictive performance in terms of $R^2$ between the full and the baseline model) between the STL and MTL approach. Once again, the blue histogram corresponds to the distribution of Granger causality retrieved using the STL approach, while the orange corresponds to the results of the MTL approach. The shift to the right of the orange histogram shows the larger ability of the MTL model to

reveal Granger-causality between climate and vegetation. Similar to the previous comparison, the Wilcoxon paired statistical test (Demšar, 2006) confirms that the results of the two approaches are overall statistically different (p-value $< 10^{-9}$). In summary, these findings highlight the potential of using the low-dimensional feature representation learned from the data to fulfill our final objective, which is the detection of vegetated areas holding a similar response to climate via a clustering approach.

## 3.2 Appropriate number of hydro-climatic biomes

As described in Sect. 2.5, there are multiple approaches that can be used to define the number of classes in a clustering problem. In our framework, we define the number of clusters by using a data-driven approach. In our analysis, we choose not to use information from any pre-defined number of vegetation and/or climate classes existing in the literature, since the ultimate goal is to identify land classes fully independently, and only based on the observed relationship between vegetation and climate. To this end, we rely on the definition of the number of clusters on the predictive performance of the MTL model. In Sect. 2.3, it is stated that the ASO-MTL approach shares the objective function of the CMTL method. This only holds if the number of clusters (which is a pre-defined parameter in the CMTL method) is equal to the value of the parameter $h$ in the ASO-MTL method, which is the dimensionality of the common feature space. In light of this equivalence relation, we experimented with a wide range of values for $h$ in a validation set, aiming to select the value of $h$ that maximises the model performance in terms of $R^2$. As such, we conclude that the best predictive performance occurs at $h = 11$, and that the appropriate number of biomes in the clustering phase equals to 11 – see Sect. S2 of the Supplementary material for more details.

The results of this hierarchical clustering (with Euclidean distance) can be visualised in a dendrogram representation, which provides an indication about the optimal number of clusters that emerge from the data set. Figure 4b depicts the dendrogram formed by our framework, with the vertical cutting line separating the data into 11 clusters. This representation allows for a visual inspection of whether the choice of 11 clusters is in line with the dissimilarities existing in the observations. As one can observe, our choice is reasonable, since the clusters at this point are quite dissimilar, based on the Euclidean distance metric, compared to hypothesized cutting lines either before or after this point. In other words, the branches of the dendrogram are already quite long at 11 clusters, indicating high dissimilarities between the resulting classes.

## 3.3 Hydro-climatic biomes

The final objective of this study is to uncover the regions in which vegetation responds in a analogous way to climate anomalies, here referred to as 'hydro-climatic biomes'. In the previous section, we investigated the appropriate number of such regions based on the information contained in our database. Figure 4a illustrates the spatial distribution of the emerging global hydro-climatic biomes. The colours depicted correspond to those of the clusters in the dendrogram of Fig. 4b. Further analysis of this dendrogram, in combination with the spatial distribution of the clusters in Fig. 4a, shows that our framework can clearly differentiate the bio-climatic behaviour of northern latitude ecosystems from those in mid- and southern latitudes. The behaviour of tropical ecoregions is unsurprisingly closer to the behaviour of sub-tropical ones, while boreal regions sharing the exposure to low temperature anomalies have a more coherent response to one another, forming the second main branch of the dendrogram. Bearing in mind the results of the Granger causality approach by Papagiannopoulou et al. (2017b), as well

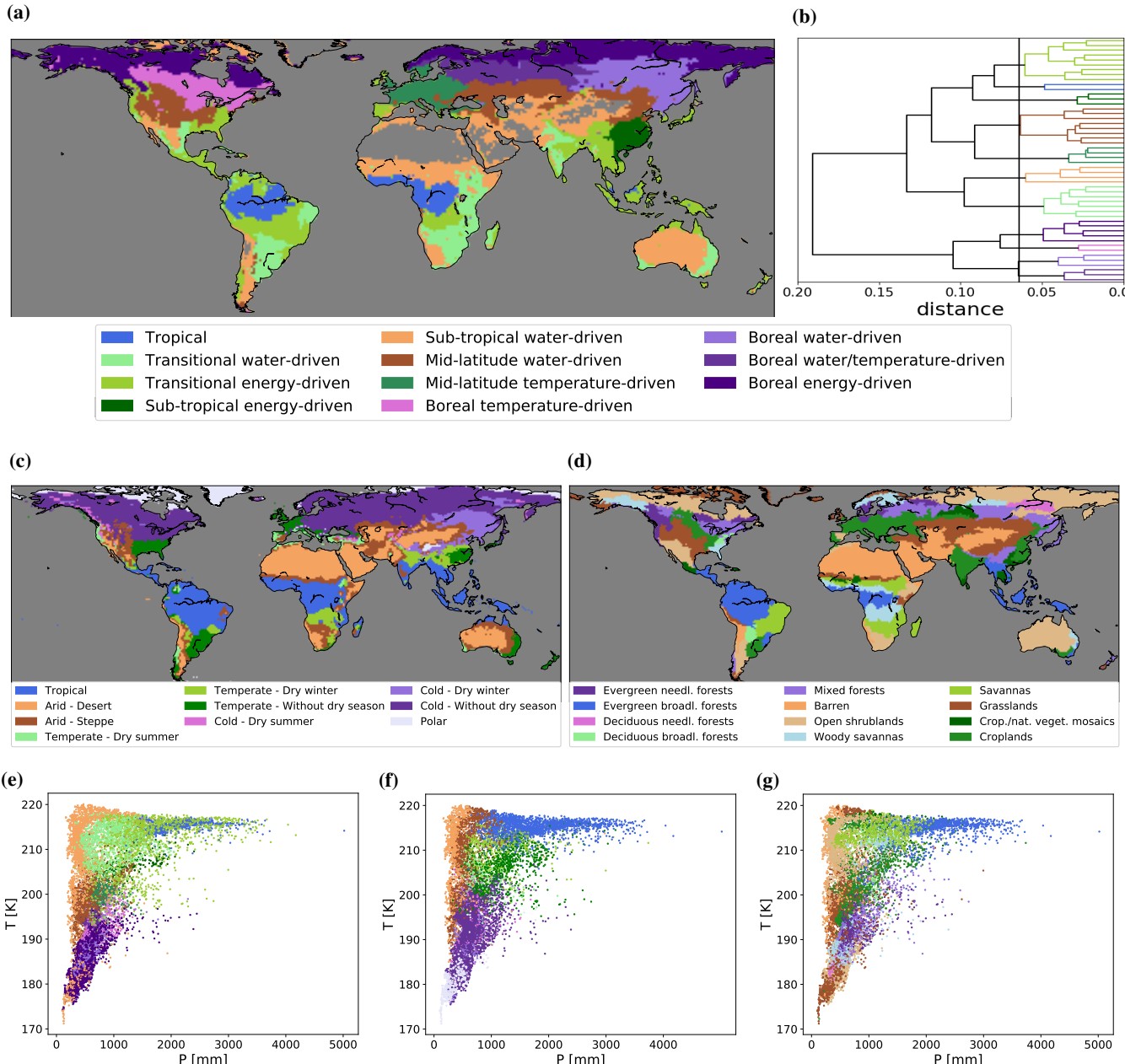

**Figure 4.** Comparison of the different land surface classification schemes. (a) Hydro-climatic biomes derived from the proposed framework. The region colours correspond to the colours of the clusters that are depicted in the dendrogram. (b) Dendrogram scheme of the clustering derived from the hierarchical agglomerative clustering on the low-dimensional representation of our model observations. The length of the dendrogram branches is a function of the inter-cluster dissimilarities. The vertical cutting line marks the data split into 11 clusters. The denomination of the different classes is supported by the results from Papagiannopoulou et al. (2017b). (c) Simplified Köppen-Geiger climate classification scheme. (d) IGBP land use classification scheme. (e) Climate space (i.e. mean annual temperature versus precipitation) for our hydro-climatic biomes in Fig. 4a. (f) Same as (e) but for the Köppen-Geiger climate classes in Fig. 4c. (g) Same as (e) but for IGBP in Fig. 4d.

as the prior knowledge on climate and land use classification, we define the hydro-climatic biomes as follows: (1) Tropical, (2) Transitional water-driven, (3) Transitional energy-driven, (4) Sub-tropical energy-driven, (5) Sub-tropical water-driven, (6) Mid-latitude water-driven, (7) Mid-latitude temperature-driven, (8) Boreal temperature-driven, (9) Boreal water-driven, (10) Boreal water/temperature-driven, (11) Boreal energy-driven. This nomenclature is broadly based on latitude and main climatic drivers.

Figure 4c shows the main 10 climate regions of the Köppen-Geiger climate classification, which is based on precipitation and temperature, and their seasonality. On the other hand, the International Geosphere-Biosphere Program (IGBP) (Loveland and Belward, 1997) land cover classification, depicted in Fig. 4d, is mostly based on plant functional types. Without the need to prescribe any land cover or climate classification, and only relying on the spatial coherence in the vegetation response to climate anomalies, our hydro-climatic biomes in Fig. 4a clearly depict some of the main characteristic patterns from these traditional classification schemes. For instance, the region of North Asia is quite coherent in terms of climate based on the 10 climate classes shown here (Fig. 4c), but quite diverse in terms of vegetation type (Fig. 4d); the hydro-climatic biomes show a clear distinction in the transition from shrublands (energy-driven) to coniferous forests (energy- and water-driven). In North America, the more energy-limited ecosystems along the coasts emerge from the water-driven regions inland, and a latitudinal behaviour is also depicted, partly reflecting the transition from croplands and grasslands into temperate and boreal forests. Patterns in the tropics clearly differentiate between rainforest and transitional savannahs, and in South America the different drivers of vegetation dynamics in the Arc of Deforestation lead to a class change that is not depicted by neither the Köppen-Geiger climate classification nor the IGBP land cover classes. Finally the patterns found for arid and warm semiarid regions (here referred to as 'sub-tropical water-driven'), and their transition towards wetter and more vegetated ecosystems, agree with the expectations based on vegetation (Fig. 4d) and climate (Fig. 4c).

The comparison to the Köppen-Geiger and IGBP maps serves only as a general evaluation or proof of concept for our hydro-climatic biomes map, since in the end such maps are based on a different rationale, and thus, there is no intent to 'outperform' these classification schemes. However, it can be observed in this comparison that the hydro-climatic biomes map in Fig. 4a combines information on climate and vegetation zones by illustrating regions where vegetation similarly interacts with the multi-month dynamics in climatic and environmental conditions. This conclusion is confirmed by the scatter plots in Figs. 4e-g. Figure 4e depicts our hydro-climatic biomes of Fig. 4a in climate space of mean annual temperature against precipitation, while Fig. 4f shows the same but for the Köppen-Geiger climate classes of Fig. 4c. In Fig. 4f, the five climate classes are well-separated, since their definition is based on these two climatic variables. On the other hand, Fig. 4g depicts the same information but for the IGBP map of Fig. 4d. In this figure, savannahs, tropics, and shrublands appear again well clustered. It can be observed that the scatter plot of Fig. 4e clearly lie between the two previous classifications in terms of clustering. Boreal biomes correspond to cold climate classes, the sub-tropical and mid-latitude water-driven biomes correspond to arid regions, while the transitional biomes correspond to the savannas and croplands. The clustering of biomes is also consistent with the global distribution of key climatic drivers reported by Papagiannopoulou et al. (2017b) based on random forests and a Granger-causality framework, since these biomes are ultimately defined based on the response of vegetation to climatic and environmental conditions. These common dynamics are identified by latent structures in our MTL approach; a discussion

on these latent structures is included in the Supplementary material (Sect. S3). Moreover, we should note that the approach of spectral clustering applied here, allows for a robust result, as small perturbations in the data sets do not affect the overall clustering result. This conclusion is confirmed by the fact that even in tropical regions, where the uncertainty in the observations is typically larger and the skill of the predictions is lower (see Fig. 3), the different clusters are separated in a clear manner.

A discussion about the comparison of the three land surface classification schemes (the hydro-climatic biomes, the Köppen-Geiger climate classification and the IGBP land use classification) is presented in Sect. S4 of the Supplementary material. Results for microwave vegetation optical depth (VOD) (Liu et al., 2011) anomalies as alternative to NDVI anomalies are consistent as shown in Supplementary material Fig. S7.

## 4   Conclusion

In this paper we introduced a novel framework for identifying regions with similar biosphere-climate dynamics interplay. Our framework combines a multi-task learning (MTL) modelling approach and a spectral clustering technique, and it is applied to a global database of global observational climate records compiled by Papagiannopoulou et al. (2017a). Comparisons to a typical single-task learning approach, in which each task (in each location) is analysed separately, indicate that learning about climate–vegetation relationships in neighbouring, or even remote, locations can help predict local vegetation dynamics based on climate variability. Moreover, our approach is able to detect shared hidden predictive structures among the tasks that enhance the performance of the models. These predictive structures form the basis to which the clustering algorithm is applied to detect regions where vegetation responds to climate in a similar way. We demonstrate that, without the need to prescribe any land cover information, our method is able to identify coherent climate–vegetation interaction zones that emerge directly from the spatio-temporal variability in the data. These zones agree with traditional global classification maps, such as the Köppen-Geiger climate classification or the IGBP land cover classification. We refer to these regions as 'hydro-climatic biomes'. These wide regions can be used in various applications in geosciences, such as unravelling anomalous relationships between climate and vegetation dynamics at local scales, defining extreme values of vegetation response to climate, exploring tipping points and turning points (Horion et al., 2016) of ecosystem resilience, and benchmarking the dynamic response of vegetation in Earth system models.

*Code and data availability.*   We use the implementation of Python for the L-BFGS optimizer, the singular value decomposition method and the hierarchical clustering (scikit-learn python library (Pedregosa et al., 2011)). The code for the ASO-MTL method (doi: 10.5281/zen-odo.1241047) has been uploaded to our github repository (https://github.com/lhwm/hydro-climatic-biomes). Data used in this manuscript can be accessed using http://www.SAT-EX.ugent.be as gateway.

*Author contributions.* Christina Papagiannopoulou, Willem Waegeman and Diego G. Miralles conceived the study. Christina Papagiannopoulou conducted the analysis. Christina Papagiannopoulou, Diego G. Miralles and Matthias Demuzere led the writing. All co-authors contributed to the design of the experiments, discussion and interpretation of results, and editing of the manuscript.

*Acknowledgements.* This work is funded by the Belgian Science Policy Office (BELSPO) in the framework of the STEREO III programme,
5    project SAT-EX (SR/00/306). D. G. Miralles acknowledges support from the European Research Council (ERC) under grant agreement no. 715254 (DRY-2-DRY). The authors thank Stijn Decubber for fruitful discussions. The authors also sincerely thank the individual developers of the wide range of global data sets used in this study. Finally, the authors thank the reviewers for their constructive feedback.

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
