# Peer review of "Global hydro-climatic biomes identified via multi-task learning"

_Geoscientific Model Development, 2018_

## Short Comment (SC1) · 4 May 2018

Following the guideline in https://www.geoscientific-model-development.net/about/manuscript_types.html model code must be available when the manuscript is submitted. As it is planned that the code is published in github this should be completed prior to submission/publication with the relevant software version tagged as a release. The authors are also encouraged to create a DOI via Zenodo, see https://guides.github.com/activities/citable-code/. The relevant data should supplied as supplement or cited via a DOI.

Lutz Gross GMD Executive Editor

---

## Author Comment (AC1) · 4 May 2018

We thank the editor for the reminder. The code and the data are already available at the provided webpages. The corresponding DOI and version of the code are the following: Version v1.0 and DOI: 10.5281/zenodo.1241047

---

## Short Comment (SC2) · 1 Jun 2018

The work presented by Papagiannopoulou et al. in this manuscript is of interest for the reader of GMD and is also very relevant for the ecosystem and climate research community. Overall the manuscript is well structured and the methodology section generally well documented. Knowing that the focus of GMD is on the progress and novelty in computation and model development, I support the need for in-depth description of the MTL model and its performances (e.g. SLT vs MLT, capability to detect Granger causality, etc.). However I believe that the manuscript would be strengthened and results better supported if the authors could really demonstrate that the new product (i.e. map of hydro-climatic biomes) is outperforming other bioclimatic maps that did not consider in their design the vegetation response to climate variability.

[Figure]

This is still lacking in the current manuscript. In addition some methodological aspects that led to the final design of the MLT and clustering should also be improved to backup the authors' statement on the performances of the final models and derived product. Based on these observations and on the detailed comments provided in the joint report I recommend the paper for major revision.

Please also note the supplement to this comment:
https://www.geosci-model-dev-discuss.net/gmd-2018-92/gmd-2018-92-SC2-supplement.pdf

**Supplement:**

**Review of gmd-2018-92**

The work presented by Papagiannopoulou et al. in this manuscript is of interest for the reader of GMD and is also very relevant for the ecosystem and climate research community. Overall the manuscript is well structured and the methodology section generally well documented. Knowing that the focus of GMD is on the progress and novelty in computation and model development, I support the need for in-depth description of the MTL model and its performances (e.g. SLT vs MLT, capability to detect Granger causality, etc.). However I believe that the manuscript would be strengthened and results better supported if the authors could really demonstrate that the new product (i.e. map of hydro-climatic biomes) is outperforming other bioclimatic maps that did not consider in their design the vegetation response to climate variability. This is still lacking in the current manuscript. In addition some methodological aspects that led to the final design of the MLT and clustering should also be improved to backup the authors' statement on the performances of the final models and derived product. Based on these observations and on the detailed comments provided below I recommend the paper for major revision.

**Specific comments**

**Introduction:**

- Studying vegetation response to climate variability is and has been the focus of numerous researches. I know the objective of the authors is to create a new bio-climatic map, however I can imagine that their work build up on recent developments in science regarding ecosystem response to climate variability. This is not well reflected in the introduction. Please add some references to key papers, studies in the matter. Some suggestions below:

Liu L, Zhang Y, Wu S, Li S, Qin D (2018) Water memory effects and their impacts on global vegetation productivity and resilience. Sci Rep, 8, 2962.

Seddon AW, Macias-Fauria M, Long PR, Benz D, Willis KJ (2016) Sensitivity of global terrestrial ecosystems to climate variability. Nature, 531, 229-232.

De Keersmaecker W, Lhermitte S, Tits L, Honnay O, Somers B, Coppin P (2015) A model quantifying global vegetation resistance and resilience to short-term climate anomalies and their relationship with vegetation cover. Global Ecology and Biogeography, 24, 539-548.

Nemani RR, Keeling CD, Hashimoto H et al. (2003) Climate-driven increases in global terrestrial net primary production from 1982 to 1999. Science, 300, 1560-1563.

- The authors claim (p2, l23) that it is the first time that ecoregions are being defined based on the analysis of vegetation response to climate variability. I agree that the idea is relatively novel and definitely relevant. Yet previous attempts have been made, notably by combining PCA and clustering techniques applied to climate and vegetation dataset. See the following reference as an example: Ivits E, Horion S, Fensholt R, Cherlet M (2014) Global Ecosystem Response Types Derived from the Standardized Precipitation Evapotranspiration Index and FPAR3g Series. Remote Sensing, 6, 4266-4288.

**Methodology**

- **Sect. 2.4.** The authors mentioned that the ASO method used here should not be confused with PCA. It would be useful to develop this statement. Indeed for both techniques orthonormal vectors are derived from the high dimensional feature space, creating a new 'optimized' low-dimensional feature space. The authors mentioned that the goal of the ASO method is to detect the PC of the predictive structure. Knowing that PCA can be performed in two ways (t-mode and s-mode), the t-mode being the most frequently used by climatologist to identify recurrent spatial patterns over time, whereas the S-mode allows for identifying recurrent temporal patterns over space. How would the current method differ from an extended PCA in S-mode? I can imagine that using EPCA over a dataset as large as the one used here could be a real challenge for example. But I would like the authors to elaborate on the pros and cons of the new method as compared to already established techniques in the climate research such as PCA/EPCA for example.
- **Sect. 2.5.** The authors do not give any name or reference for the clustering technique used here. Please clarify if a new algorithm has been developed for the study or if an already developed clustering technique was applied.
- **General comment on the use of R2 for assessing the model performance**: at several occasions (in the manuscript and in the supplementary material), the authors used r2 to quantify the performance of different models (MLT vs. SLT, models with and without Granger causality, inclusion of higher-level features in the input dataset, final decision on the number of clusters). They generally conclude that the best model is the one with the highest r2. I agree on the principle, however looking at the differences between r2 (e.g. figures 3b and 3d, large areas present difference in r2 below 0.1), I wonder whether all these differences are statistically significant. As based on the analysis of r2, the authors are deciding on the final set of input data, the final design of the MLT model, and the final number of clusters, I would really urge the need for further statistical assessment of the model performances. One first analysis could simply be to estimate the percentage area of pixels with statistically significant increase in r2.

**Results:**

- General comment on the final number of clusters: the fact that the majority of the Iberian Peninsula is included in the transitional energy driven cluster together with Ireland, an important part of SE Asia, part of Brasil and Venezuela – Colombia makes me wonder if a higher number of clusters would not be more appropriate. The authors mentioned already in Figure S2 that the differences in the predictive performance for h= 6 - 15 are marginal. Further assessments should therefore be performed in order to identify the optimal number of hydro-climatic biomes. Part of this assessment should be dedicated to the understanding of the actual drivers (main predictors) for each biome. I believe providing a solid justification for the naming of the different biomes (by referring back to the main predictors) would be beneficial for the paper.
- In relation to the previous comment, how does the new global map of hydro-climatic biomes perform as compared to previous ones (not including information of vegetation condition and response to climate)? It would be really interesting if the authors could showcase for one (or more) bio-climatic

zone how the new bio-climatic zone provide a finer, more accurate picture of global terrestrial biomes by analysis the specific (/sub-local) ecosystem response to climate variability. To this regard, the bio-climatic map produced by Metzger et al. (see reference below) could also be of interest for comparison.

Metzger MJ, Bunce RGH, Jongman RHG et al. (2012) A high-resolution bioclimate map of the world: a unifying framework for global biodiversity research and monitoring. Global Ecology and Biogeography, 22, 630–638.

- Figure 4. (c) The Koeppen classification divides the world into 5 main classes and 29 sub-classes. The authors should justify the use of 10 classes in the figure. This can be very misleading when looking and interpreting the results. An example: I do not think that the statement p14, l21-23 '… the region of North Asia is coherent in terms of climate, but quite diverse in terms of vegetation types ; the hydro-climatic biomes show a clear distinction from shrublands (…) to coniferous …' holds entirely when looking at the high level details (29 classes) of the Koeppen classification. Please justify your choice here.

- Supplementary material S4. The authors indicate that the best-formed clusters are depicted in FigS4a (hence by the hydro-climatic biomes). I find very difficult to make any final judgment of the best "depiction" (/detection) of biomes based on the 2dimensional graphs provided here.

**Technical comments**

- P5, l14: please add a reference for the statement: '… this kind of modelling is becoming more common in climate science…'

- P10, l10: please clarify what you mean by multi-month vegetation dynamics. Is it seasonal, sub-seasonal, yearly?

- P12, l5: please correct 'Geanger' with 'Granger'

- Figure 4. (a) the color code for the clusters sub-tropical energy driven and mid-latitude temperature driven are too similar. It is difficult to differentiate them. Please adjust the color scheme of the legend.

- p15, l22: The term 'turning point' has only been introduced recently in ecosystem and climate science so for clarity, you can refer to:

Horion S, Prishchepov AV, Verbesselt J, De Beurs K, Tagesson T, Fensholt R (2016) Revealing turning points in ecosystem functioning over the Northern Eurasian agricultural frontier. Glob Chang Biol, 22, 2801-2817.

---

## Author Comment (AC2) · 18 Jun 2018

**Response to the review comments of Stephanie Horion**

The work presented by Papagiannopoulou et al. in this manuscript is of interest for the reader of GMD and is also very relevant for the ecosystem and climate research community. Overall the manuscript is well structured and the methodology section generally well documented. Knowing that the focus of GMD is on the progress and novelty in computation and model development, I support the need for in-depth description of the MTL model and its performances (e.g. STL vs MTL, capability to detect Granger causality, etc.). However I believe that the manuscript would be strengthened and results better supported if the authors could really demonstrate that

the new product (i.e. map of hydro-climatic biomes) is outperforming other bio-climatic maps that did not consider in their design the vegetation response to climate variability. This is still lacking in the current manuscript. In addition some methodological aspects that led to the final design of the MTL and clustering should also be improved to backup the authors' statement on the performances of the final models and derived product. Based on these observations and on the detailed comments provided below I recommend the paper for major revision.

**We would like to thank the reviewer for her appreciation of our manuscript, the constructive feedback, and thorough assessment. Below we provide a point-by-point response to each comment.**

**In general, we would like to clarify that the goal of our study is to provide a new methodology that can identify coherent regions in which vegetation responds to climate in a similar way. We model our problem with a multi-task learning approach that considers the different locations as different tasks and learns the relationship between the tasks during the learning process. Hence, the climate–vegetation interaction is simultaneously learned for all locations. The similarity between the learned relationships (between the tasks) is also discovered during the process. This is the first time (to the best of our knowledge) that an approach of this kind, which discovers the structure of the relationships between the different locations, is applied on this setting. As such, we try to avoid the claim that our *hydro-climatic biomes* 'outperform' other schemes, which rely on climate and/or vegetation data and not on the modeled interaction between climate and vegetation. It is not really our intent to outperform these land cover classifications, and the comparison that is provided against them is to assure that – despite the fact that our approach does not prescribe any explicit information on land cover types – comparable regions arise from our data-guided appraisal.**

**Specific comments**

Introduction
- Studying vegetation response to climate variability is and has been the focus of numerous researches. I know the objective of the authors is to create a new bio-climatic map, however I can imagine that their work build up on recent developments in science regarding ecosystem response to climate variability. This is not well reflected in the introduction. Please add some references to key papers, studies in the matter. Some suggestions below:

Liu L, Zhang Y, Wu S, Li S, Qin D (2018) Water memory effects and their impacts on global vegetation productivity and resilience. Sci Rep, 8, 2962.

Seddon AW, Macias-Fauria M, Long PR, Benz D, Willis KJ (2016) Sensitivity of global terrestrial ecosystems to climate variability. Nature, 531, 229-232.

De Keersmaecker W, Lhermitte S, Tits L, Honnay O, Somers B, Coppin P (2015) A model quantifying global vegetation resistance and resilience to short-term climate anomalies and their relationship with vegetation cover. Global Ecology and Biogeography, 24, 539-548.

Nemani RR, Keeling CD, Hashimoto H et al. (2003) Climate-driven increases in global terrestrial net primary production from 1982 to 1999. Science, 300, 1560-1563

**True. We will include the suggested literature in the revised manuscript.**

- The authors claim (p2, l23) that it is the first time that ecoregions are being defined based on the analysis of vegetation response to climate variability. I agree that the idea is relatively novel and definitely relevant. Yet previous attempts have been made, notably by combining PCA and clustering techniques applied to climate and vegetation dataset. See the following reference as an example:

Ivits E, Horion S, Fensholt R, Cherlet M (2014) Global Ecosystem Response Types

Derived from the Standardized Precipitation Evapotranspiration Index and FPAR3g Series. Remote Sensing, 6, 4266-4288.

**Thanks for pointing us to this paper. The suggested literature is relevant to our study and will also be referred to in the revised version. In addition, the differences compared to this and other studies will also be highlighted in the revised version.**

Methodology

- Sect. 2.4. The authors mentioned that the ASO method used here should not be confused with PCA. It would be useful to develop this statement. Indeed for both techniques orthonormal vectors are derived from the high dimensional feature space, creating a new 'optimized' low-dimensional feature space. The authors mentioned that the goal of the ASO method is to detect the PC of the predictive structure. Knowing that PCA can be performed in two ways (t-mode and s-mode), the t-mode being the most frequently used by climatologist to identify recurrent spatial patterns over time, whereas the S-mode allows for identifying recurrent temporal patterns over space. How would the current method differ from an extended PCA in S-mode? I can imagine that using EPCA over a dataset as large as the one used here could be a real challenge for example. But I would like the authors to elaborate on the pros and cons of the new method as compared to already established techniques in the climate research such as PCA/EPCA for example.

**Thanks for this comment. In the last paragraph of Sect. 2.4, we mention the main difference between the commonly-used PCA approaches and the proposed method. However, we will elaborate on the differences and potential advantages of our approach in the revised manuscript.**

**To give an example, in the work of Ivits et al. (2014), PCA is performed on the data**

matrix of drought anomalies (measured by Standard Precipitation Evapotranspiration Index data, SPEI) and vegetation state (measured by Fraction of Photosynthetically Active Radiation data, FPAR3g), while the clustering is applied to the correlation coefficients based on the spatio-temporal patterns obtained by PCA.

Our approach is based on different principles, and as such it is expected to yield different results. We explicitly consider the climatic variables as predictors and the vegetation variable as target variable, and we learn the relationship between them in a supervised setting. As such, the regions that we define rely on the relationship between climate and vegetation in a prediction setting, and the clustering is calculated based on similarity of this relationship (i.e. the model coefficients for different locations). As such, we learn relationships between climate and vegetation in a supervised setting, whereas PCA-based methods are fully unsupervised. In our study the SVD decomposition is used as part of the optimization algorithm, thus in a supervised setting. In this setting, the model weights are optimized based on a given training set. Therefore, the discovered structures are obtained during the training process. This novel part of our methodology will be stated more clear in the revised version.

- Sect. 2.5. The authors do not give any name or reference for the clustering technique used here. Please clarify if a new algorithm has been developed for the study or if an already developed clustering technique was applied.

In the manuscript, it is mentioned that the clustering technique that we use is the agglomerative hierarchical clustering (with Euclidean distance measure) which is a well-known clustering method in Statistics (see Sect. 2.5 and 3.2 of the manuscript). To make it more clear to the broad audience of GMD, we will mention in the revised manuscript that we use the hierarchical clustering python implementation of scikit-learn, and add a specific reference.

- General comment on the use of $R^2$ for assessing the model performance: at several occasions (in the manuscript and in the supplementary material), the authors used $R^2$ to quantify the performance of different models (MTL vs. STL, models with and without Granger causality, inclusion of higher-level features in the input dataset, final decision on the number of clusters). They generally conclude that the best model is the one with the highest $R^2$. I agree on the principle, however looking at the differences between $R^2$ (e.g. figures 3b and 3d, large areas present difference in $R^2$ below 0.1), I wonder whether all these differences are statistically significant. As based on the analysis of $R^2$, the authors are deciding on the final set of input data, the final design of the MTL model, and the final number of clusters, I would really urge the need for further statistical assessment of the model performances. One first analysis could simply be to estimate the percentage area of pixels with statistically significant increase in $R^2$.

**The distributions depicted in Figs. 3c and 3f of the manuscript show that the results of the MTL and STL methods are substantially different. Specifically, the distributions of the MTL results are shifted to the right, meaning that STL is outperformed by MTL at global scale. This result can be confirmed by any paired statistical test (Demšar, 2006). The same comparison can be applied for the performance comparison of the full and the baseline MTL models.**

**However, we agree that a significance test of this difference was not included in the original manuscript. At pixel level, traditional statistical tests usually have too many assumptions for our purposes. Alternatively, non-parametric tests based on resampling, such as permutation or bootstrap tests, cannot really be applied due to the size of our data set. A proposed solution is to use the Diebold-Mariano statistical test (Diebold, 2015). This test can be used here to compare the MTL and the STL approaches and will be used in the revised version of the manuscript.**
**For the final decision about the number of clusters, see our answer below.**

Results

- General comment on the final number of clusters: the fact that the majority of the Iberian Peninsula is included in the transitional energy driven cluster together with Ireland, an important part of SE Asia, part of Brasil and Venezuela – Colombia makes me wonder if a higher number of clusters would not be more appropriate. The authors mentioned already in Figure S2 that the differences in the predictive performance for h = 6 - 15 are marginal. Further assessments should therefore be performed in order to identify the optimal number of hydro-climatic biomes. Part of this assessment should be dedicated to the understanding of the actual drivers (main predictors) for each biome. I believe providing a solid justification for the naming of the different biomes (by referring back to the main predictors) would be beneficial for the paper.

**We agree that the differences in predictive performance for h = 6-15 are marginal. However, the proposed method is a fully data-driven approach that is not fine-tuned based on any kind of prior knowledge. Therefore, the selection of the final value of the h parameter is based on an objective criterion, i.e. the model performance. As for the resulting map (Fig. 4a), although we are aware that this map may not fully reflect all particular expectations, we do believe that the spatial distribution broadly captures the expected regimes of climate–vegetation interactions, as described in the results section. Note as well that in our early experiments we ran our approach with a different number of clusters to visually inspect the resulting regions. The regions formed with h values close to 11 are similar to the reported ones (Fig. 4a of the manuscript). This result proves the robustness of the proposed method to detect the basic vegetation response types with respect to climate. These results (for h = 8-12) will be included as supplementary figures in the revised manuscript.**

**Concerning the labels scheme, we should stress that the names of the biomes are inspired by the main predictors based on Papagiannopoulou et al. (2017). We are afraid that making the labels reflect these predictors more accurately would make them extremely complex and rather impractical.**

- In relation to the previous comment, how does the new global map of hydro-climatic biomes perform as compared to previous ones (not including information of vegetation condition and response to climate)? It would be really interesting if the authors could showcase for one (or more) bio-climatic zone how the new bio-climatic zone provide a finer, more accurate picture of global terrestrial biomes by analysis the specific (/sub-local) ecosystem response to climate variability. To this regard, the bioclimatic map produced by Metzger et al. (see reference below) could also be of interest for comparison.
Metzger MJ, Bunce RGH, Jongman RHG et al. (2012) A high-resolution bioclimate map of the world: a unifying framework for global biodiversity research and monitoring. Global Ecology and Biogeography, 22, 630-638.

**Thanks for the relevant reference, it will definitely be cited in the revised manuscript. However, as we mentioned above, by using our approach we really aim for detecting regions of consistent behavior in response to climate (based on the learned weights). That is what we should evaluate. As such, we cannot really aim for 'accurate' biomes. This is the reason why we do not compare our result to other data-driven approaches that rely on climate and/or vegetation data (as Metzger et al. (2012)), since our study tries to detect regions based on different criteria (based on the interaction between climate–vegetation and not on the data). This point will also be stressed in the revised manuscript. We also note again that the comparison that is provided against traditional land classification schemes is to assure that comparable regions arise from our data-guided approach, despite these land cover types not being expecificaly**

**prescribed.**

- Figure 4. (c) The Köppen classification divides the world into 5 main classes and 29 sub-classes. The authors should justify the use of 10 classes in the figure. This can be very misleading when looking and interpreting the results. An example: I do not think that the statement p14, l21-23 '...the region of North Asia is coherent in terms of climate, but quite diverse in terms of vegetation types; the hydro-climatic biomes show a clear distinction from shrublands (...) to coniferous ...' holds entirely when looking at the high level details (29 classes) of the Köppen classification. Please justify your choice here.

**It is true that the Köppen climate classification scheme consists of divisions and sub-divisions of the five main climate types. We could choose to use the divisions of the Köppen classification, which are basically 12 (if we also divide the tropics further) and not 10 as in Fig. 4. However, the use of 10 instead of 12 classes will not make the map look much different. Moreover, from the color scheme used in Fig. 4, it is clear that there are five main classes. In Fig. 4, we aim for comparing the regions detected by the proposed method to the regions based on the Köppen climate classification scheme. Since the division of 10 climate classes is closer to the number of regions detected by our approach, we choose this number of regions (10) on Köppen's map. Nonetheless, we agree that the statements mentioned in the comment sound a bit strong, so in our discussion we should take into account also the sub-division (29 classes) of Köppen classification. Again, the comparison to the Köppen and IGBP maps serves only as a general evaluation or proof of concept for our hydro-climatic biomes map, since in the end they are based on a different rationale. Thus, we will clarify in the revised manuscript that we do not claim that our map is capable of 'outperforming' these classification schemes.**

- Supplementary material S4. The authors indicate that the best-formed clusters are depicted in FigS4a (hence by the hydro-climatic biomes). I find very difficult to make any final judgment of the best "depiction" (/detection) of biomes based on the 2-dimensional graphs provided here.

**Yes, true. Another dimensionality reduction technique, such as the t-sne, might give a visually better result. We are exploring the potential of other methods in order to improve these figures in the revised version.**

Technical comments

- P5, l14: please add a reference for the statement: '...this kind of modelling is becoming more common in climate science...'

**This sentence refers to the previously mentioned studies, which are described in the same paragraph, and serves as a conclusion that MTL approaches are used more common recently than in the past in climate science. However, we will repeat the references at this point as well in the revised manuscript.**

- P10, l10: please clarify what you mean by multi-month vegetation dynamics. Is it seasonal, subseasonal, yearly?

**We mean monthly vegetation. It will be corrected.**

- P12, l5: please correct 'Geanger' with 'Granger'

**True, thanks.**

- Figure 4. (a) the color code for the clusters sub-tropical energy driven and

mid-latitude temperature driven are too similar. It is difficult to differentiate them. Please adjust the color scheme of the legend.

**Indeed. We will adjust the color scheme in the revised manuscript.**

- p15, l22: The term 'turning point' has only been introduced recently in ecosystem and climate science so for clarity, you can refer to:
Horion S, Prishchepov AV, Verbesselt J, De Beurs K, Tagesson T, Fensholt R (2016) Revealing turning points in ecosystem functioning over the Northern Eurasian agricultural frontier. Glob Chang Biol, 22, 2801-2817.

**We will include this relevant reference in the revised manuscript.**

References

Demšar, J. Statistical comparisons of classifiers over multiple data sets. Journal of Machine learning research. 2006, 7(Jan), 1-30.

Diebold, F. X. Comparing predictive accuracy, twenty years later: A personal perspective on the use and abuse of Diebold-Mariano tests. 2015, J. Bus. Econ. Stat., 33.

Papagiannopoulou, C., Miralles, D. G., Dorigo, W. A., Verhoest, N. E. C., Depoorter, M., and Waegeman, W. Vegetation anomalies caused by antecedent precipitation in most of the world, Environ. Res. Lett., 2017, doi:10.1088/1748-9326/aa7145.

Ivits, E., Horion, S., Fensholt, R., Cherlet, M. Global Ecosystem Response Types Derived from the Standardized Precipitation Evapotranspiration Index and FPAR3g

Series. Remote Sensing, 2014, 6, 4266-4288.

Metzger, M. J., Bunce, R. G. H., Jongman, R. H. G. et al. A high-resolution bioclimate map of the world: a unifying framework for global biodiversity research and monitoring. Global Ecology and Biogeography, 2012, 22, 630-638.

---

## Referee Comment (RC2) · Anonymous Referee #2 · 4 Jul 2018

This study presents an new approach for the classification of global biomes. The idea is to focus on the statistical sensitivities of NDVI anomalies to multiple predictors. I do think that it is important to emphasize the "goal" of classification, and therefore the paper is a step in the right direction. I have, however, doubts if focusing on NDVI anomalies is the right target. In particular for tropical ecosystems NDVI does not tell us much about ecosystem dynamics and the figures show the underlying predictions are indeed not convincing. Hence, I have some doubts about the novelty that this classification can offer. Similar as all classical approaches, also this method fails to reveal the complex spatial patterns in tropical ecosystems. This is why I see this paper more as a methodological contribution that can actually help future studies to realize analogous exercises based on different data sets.

[Figure]

Overall, the approach of the paper is to stack a series of methods. First, "Multi-Task Learning" is used to create a statistical prediction model whose sensitivities (condensed by SVD) later serve as basis for clustering. I applaud the authors for identifying a machine learning method that seems to captures spatial relationships. But my question is if there is no corresponding geostatistical approach out there that could be equally used (e.g. a GWR or so) which deals exactly with such questions? In particular, I believe (but don't know) that the MTL does not consider the fact that lat-lon grid cells represent different geographical distances, or how do the authors considered that a global analysis is executed on a sphere?

The paper is neatly written, but I still had trouble finding my way through the paper. One aspect is that it is difficult to follow the paper without knowing the author's previous papers. In addition, I spent most of my time understanding Multi Task Learning. In particular section 2.4. was hard to understand. At this crucial point I would ask the authors to consider rewriting the paper in a way that can be understood intuitively by environmental scientists who are not familiar with the method. Likewise the link to clustering is a bit opaque. What is a "hierarchical agglomerative clustering approach"? Etc.

What irritated me about the results is that the prediction method does not manage to explain more than 40% of the variance (why else would the scale in Fig. 3 a otherwise be cut off at $\geq 0.4$?). This is actually a bit disappointing and suggests that the regression model was not the right choice, or?

Minor remarks:

The introduction does not provide a systematic overview of alternative approaches. Rather, we find here a rather random selection of climate and land cover classifications and the wording is not always correct. For example, the paper speaks of "big data" approaches, but I did not find any of the referenced studies really dealing with big data topics ("volume", "diversity", "speed", ...). I think we are talking here about (sometimes semi-heuristic), but essentially classical data exploration and machine learning methods. So, I think it would be nice to revise this part a bit to have a smooth start.

The paper is full of shortcuts such as "detrended seasonal NDVI anomalies", which are not as clear as they appear at first glance. I can think of a large number of possibilities for robustly estimating (linear/non-linear) trends and a further variety of methods for estimating seasonal cycles. It would be nice if such statements were more precise.

The same comment applies to the selection of predictors e.g. seasonal anomalies, detrended seasonal anomalies, time delayed variables, and cumulative variables etc. look like a very arbitrary selection of predictors. In a paper that has a strong affinity to data-driven methods, I would expect a more formal variable selection following a clearly defined cost function. Maybe this is too late now, but still one question can be answered: why are these predictors all regarded as "non-linear"? In most cases, they read like fairly linear transformations (maybe with the exception of cumulative variables).
* * *

---

## Author Comment (AC3) · 11 Jul 2018

**Response to Anonymous Referee#2**

This study presents a new approach for the classification of global biomes. The idea is to focus on the statistical sensitivities of NDVI anomalies to multiple predictors. I do think that it is important to emphasize the "goal" of classification, and therefore the paper is a step in the right direction.

We would like to thank the reviewer for the constructive feedback and thorough assessment. Below we provide a point-by-point response to each comment.

I have, however, doubts if focusing on NDVI anomalies is the right target. In particular for tropical ecosystems NDVI does not tell us much about ecosystem dynamics and the figures show the underlying predictions are indeed not convincing. Hence, I have some doubts about the novelty that this classification can offer. Similar as all classical approaches, also this method fails to reveal the complex spatial patterns in tropical ecosystems. This is why I see this paper more as a methodological contribution that can actually help future studies to realize analogous exercises based on different data sets.

We agree with the reviewer that although NDVI is a commonly-used index, it is known to saturate in tropical ecosystems. As we discussed in our previous work (Papagiannopoulou et al., 2017), the low predictive power of our model in tropical regions can be explained by the fact that in these regions, (i) the uncertainty in the data is larger, and (ii) vegetation might be primarily affected by other factors such as nutrient availability (rather than climate). However, with the proposed data-driven framework, pixels that belong to these tropical regions are grouped together. This means that the learned weight vectors of these pixels are similar and thus the clustering algorithm is able to detect these similarities to conform a coherent biome. Moreover, we also agree that our work can be seen as a methodological contribution, since it can be used in different application scenarios or with an alternative target variable. So, we are willing to explore the applicability of the method to a different target variable. As such, the applicability to microwave Vegetation Optical Depth (VOD) anomalies, instead of the NDVI anomalies, will be explored in the revised manuscript. VOD is known to be less sensitive to saturation in densely-vegetated biomes.

Overall, the approach of the paper is to stack a series of methods. First, "Multi-Task Learning" is used to create a statistical prediction model whose sensitivities (condensed by SVD) later serve as basis for clustering. I applaud the authors for
identifying a machine learning method that seems to capture spatial relationships. But my question is if there is no corresponding geostatistical approach out there that could be equally used (e.g. a GWR or so) which deals exactly with such questions? In particular, I believe (but don't know) that the MTL does not consider the fact that lat-lon grid cells represent different geographical distances, or how do the authors considered that a global analysis is executed on a sphere?

As we have described in the manuscript, our approach is purely data-driven. Therefore, we stress that we do not include any prior knowledge about the distances between the different pixels. On the contrary, we let the method learn the relationships between the different pixels. As such, the method may even group together remote pixels in which vegetation might have similar response to climate. Other geostatistical approaches, such as the GWR, assume that neighboring pixels have a similar behaviour with respect to specific variables. In these approaches, similarities between the pixels are learned by defining each time a single pixel as centroid and tuning the parameter of relatedness between this particular pixel and the surrounding pixels. In our work, we prefer to avoid this kind of neighborhood assumptions and focus on the discovery of relationships between the pixels based on the similarity in climate-vegetation interaction. We are also interested in methods that can be applied on large data sets with global coverage. However, we think that the suggested literature (about geostatistical approaches) is relevant to our study. We will refer to it in the revised manuscript.

The paper is neatly written, but I still had trouble finding my way through the paper. One aspect is that it is difficult to follow the paper without knowing the author's previous papers. In addition, I spent most of my time understanding Multi Task Learning. In particular section 2.4. was hard to understand. At this crucial point I would ask the authors to consider rewriting the paper in a way that can be understood intuitively by
environmental scientists who are not familiar with the method. Likewise the link to clustering is a bit opaque. What is a "hierarchical agglomerative clustering approach"? Etc.

We will expand section 2.4 to make the method more intuitive for the broad audience of GMD. Specifically, we aim to provide additional explanations for the notation used in our model. This way, environmental researchers that are not familiar with certain machine learning terminology will be able to have a better understanding of the proposed data-driven method. In the manuscript, it is mentioned that the clustering technique that we use is the agglomerative hierarchical clustering (with Euclidean distance measure) which is a well-known clustering method in Statistics (see Sect. 2.5 and 3.2 of the manuscript). As we mentioned in our response to the Referee#1, we will include in the revised manuscript that we use the hierarchical clustering python implementation of scikit-learn, and add a specific reference.

What irritated me about the results is that the prediction method does not manage to explain more than 40% of the variance (why else would the scale in Fig. 3 a otherwise be cut off at  $\geq$  0.4?). This is actually a bit disappointing and suggests that the regression model was not the right choice, or?

In our study, the seasonal cycle from the NDVI time series is removed. Therefore, the task of predicting the NDVI anomalies is more difficult than just predicting the raw NDVI time series. This is due to the fact that the presence of autocorrelation in the NDVI anomalies time series is much lower. Note that if we target the raw NDVI time series (which includes the seasonal component), the  $R^2$  is close to 1 in most of the regions (Papagiannopoulou et al., 2017). In addition, it is worth noting that there are other factors – such as fires, harvesting, etc. – that affect vegetation dynamics but are not included in the data set. Therefore, we should be aware that we focus on explaining the variance of the NDVI
**anomalies, taking into account only climatic variables.**

**Minor remarks:**

The introduction does not provide a systematic overview of alternative approaches. Rather, we find here a rather random selection of climate and land cover classifications and the wording is not always correct. For example, the paper speaks of "big data" approaches, but I did not find any of the referenced studies really dealing with big data topics ("volume", "diversity", "speed", ...). I think we are talking here about (sometimes semi-heuristic), but essentially classical data exploration and machine learning methods. So, I think it would be nice to revise this part a bit to have a smooth start.

In general, we would like to stress that the goal of our study is to provide a new data-driven methodology that can identify coherent regions in which vegetation responds to climate in a similar way. To the best of our knowledge, there are no other works that study this particular problem at global scale, with the arguable exception of the article pointed to by Referee#1 (lvits et al., 2014). In addition, in the manuscript, we describe the most naive approach that one could follow by using single-task learning techniques (and by learning one model per pixel). In the Introduction, we provide an overview of the most related works to our study that indeed use machine learning methods and/or prior knowledge. The term "big data" is used to explain that data-driven methods have been applied on climate data sets, which are inherently characterized by their volume, diversity, etc. We think that our work builds upon and goes one step further from previous efforts, such as the ones described in the Introduction, since it combines information from climate and vegetation and models the relationship between them. We will clarify these aspects in the revised manuscript and include literature related to geostatistical approaches
**used in modelling climate-vegetation interactions (see e.g. Zhao et al., 2015).**

The paper is full of shortcuts such as "detrended seasonal NDVI anomalies", which are not as clear as they appear at first glance. I can think of a large number of possibilities for robustly estimating (linear/non-linear) trends and a further variety of methods for estimating seasonal cycles. It would be nice if such statements were more precise.

We agree that these terms are not clearly described in the manuscript, and understand that the article should stand alone without the need of prior knowledge with regards to our previous work. We will add additional statements to briefly describe this terminology.

The same comment applies to the selection of predictors e.g. seasonal anomalies, detrended seasonal anomalies, time delayed variables, and cumulative variables etc. look like a very arbitrary selection of predictors. In a paper that has a strong affinity to data-driven methods, I would expect a more formal variable selection following a clearly defined cost function. Maybe this is too late now, but still one question can be answered: why are these predictors all regarded as "non-linear"? In most cases, they read like fairly linear transformations (maybe with the exception of cumulative variables)

We refer the reviewer to our previous answer for the first part of the comment. In addition, we would like to stress that our choice to use this set of predictors is based on the previous literature, as it has been analytically described in Papagiannopoulou et al. (2017). These constructed predictors are regarded as "non-linear", because their derivation from the raw data is not linear (see e.g. calculation of extreme indices).

References
Papagiannopoulou, C., Miralles, D. G., Decubber, S., Demuzere, M., Verhoest, N. E. C., Dorigo, W. A., and Waegeman, W. A non-linear Granger-causality framework to investigate climate-vegetation dynamics, Geosci. Model Dev., 2017, 10, 1945-1960

lvits, E., Horion, S., Fensholt, R., Cherlet, M. Global Ecosystem Response Types Derived from the Standardized Precipitation Evapotranspiration Index and FPAR3g Series. Remote Sensing, 2014, 6, 4266-4288.

Zhao, Z., et al. Exploring spatially variable relationships between NDVI and climatic factors in a transition zone using geographically weighted regression. Theoretical and Applied Climatology 120.3-4, 2015, 507-519.

---

## Author Response (AR1)

**Global hydro-climatic biomes via mulit-task learning**

Christina Papagiannopoulou, Diego G. Miralles, Matthias Demuzere, Niko E. C. Verhoest, Willem Waegeman

**Authors' response to the Referees**

For clarifying our answers to the reviewers' comments, the following scheme is used: comments of the reviewers are denoted in plain font, our answers are denoted in **bold** font and quotes from the manuscript are denoted in ***bold italic***. We also number the different comments of the reviewers as: I.J, where I is the number of the referee and J the number of the remark.

**Response to the comments of Stephanie Horion**

The work presented by Papagiannopoulou et al. in this manuscript is of interest for the reader of GMD and is also very relevant for the ecosystem and climate research community. Overall the manuscript is well structured and the methodology section generally well documented. Knowing that the focus of GMD is on the progress and novelty in computation and model development, I support the need for in-depth description of the MTL model and its performances (e.g. STL vs MTL, capability to detect Granger causality, etc.). However I believe that the manuscript would be strengthened and results better supported if the authors could really demonstrate that the new product (i.e. map of hydro-climatic biomes) is outperforming other bio-climatic maps that did not consider in their design the vegetation response to climate variability. This is still lacking in the current manuscript. In addition some methodological aspects that led to the final design of the MTL and clustering should also be improved to backup the authors statement on the performances of the final models and derived product. Based on these observations and on the detailed comments provided below I recommend the paper for major revision.

**We would like to thank the reviewer for her appreciation of our manuscript, the constructive feedback, and thorough assessment. Below we provide a point-by-point response to each comment.**
**In general, we would like to clarify that the goal of our study is to provide a new methodology that can identify coherent regions in which vegetation responds to climate in a similar way. We model our problem with a multi-task learning approach that considers the different locations as different tasks and learns the relationship between the tasks during the learning process. Hence, the climate–vegetation interaction is simultaneously learned for all locations. The similarity between the learned relationships (between the tasks) is also discovered during the process. This is the first time (to the best of our knowledge) that an approach of this kind, which discovers the structure of the relationships between the different locations at global**

scale, is applied on this setting. As such, we try to avoid the claim that our *hydro-climatic biomes* 'outperform' other schemes, which rely on climate and/or vegetation data and not on the modelled interaction between climate and vegetation. It is not really our intent to outperform these land cover classifications, and the comparison that is provided against them is to assure that – despite the fact that our approach does not prescribe any explicit information on land cover types – comparable regions arise from our data-guided appraisal.

**Specific comments**

Introduction
1.1) Studying vegetation response to climate variability is and has been the focus of numerous researches. I know the objective of the authors is to create a new bio-climatic map, however I can imagine that their work build up on recent developments in science regarding ecosystem response to climate variability. This is not well reflected in the introduction. Please add some references to key papers, studies in the matter. Some suggestions below:
Liu L, Zhang Y, Wu S, Li S, Qin D (2018) Water memory effects and their impacts on global vegetation productivity and resilience. Sci Rep, 8, 2962.
Seddon AW, Macias-Fauria M, Long PR, Benz D, Willis KJ (2016) Sensitivity of global terrestrial ecosystems to climate variability. Nature, 531, 229-232.
De Keersmaecker W, Lhermitte S, Tits L, Honnay O, Somers B, Coppin P (2015) A model quantifying global vegetation resistance and resilience to short-term climate anomalies and their relationship with vegetation cover. Global Ecology and Biogeography, 24, 539-548.
Nemani RR, Keeling CD, Hashimoto H et al. (2003) Climate-driven increases in global terrestrial net primary production from 1982 to 1999. Science, 300, 1560-1563

**We have included the suggested literature in the revised manuscript.**

p2.23: *Previous studies rely on spectral information, supervised techniques or clustering approaches, which are applied to observations of climate variables and/or vegetation characteristics. However, these classification schemes are not based on the type of response of vegetation to climate dynamics. Recent advances in understanding vegetation response to climate variability highlight the importance of revealing the sensitivity of ecosystems to climate conditions, see Nemani et al. (2003); De Keersmaecker et al. (2015); Seddon et al. (2016); Papagiannopoulou et al. (2017); Liu et al. (2018). Therefore, a step beyond these previous studies is a spatial characterization of the vegetation dynamics that are induced by climate variability, so that ecosystems of similar response to climate anomalies can be unveiled.*

1.2) The authors claim (p2, l23) that it is the first time that ecoregions are

being defined based on the analysis of vegetation response to climate variability. I agree that the idea is relatively novel and definitely relevant. Yet previous attempts have been made, notably by combining PCA and clustering techniques applied to climate and vegetation dataset. See the following reference as an example:

Ivits E, Horion S, Fensholt R, Cherlet M (2014) Global Ecosystem Response Types Derived from the Standardized Precipitation Evapotranspiration Index and FPAR3g Series. Remote Sensing, 6, 4266-4288.

**Thanks for pointing us to this paper. The reference is relevant to our study and has been referred to in the revised version. The differences compared to this and other studies have been also highlighted in the revised version.**

p2.34: *A previous effort towards detecting regions with similar vegetation response to climate involves the work of Ivits et al. (2014),where PCA is performed on the data matrix of drought anomalies and vegetation state, and a clustering is applied to the correlation coefficients based on the spatio-temporal patterns obtained by PCA. However, in this study, the interaction between climate and vegetation is not explicitly learned, nor the causes behind vegetation changes are inferred in a predictor-target framework.*

Methodology

1.3) Sect. 2.4. The authors mentioned that the ASO method used here should not be confused with PCA. It would be useful to develop this statement. Indeed for both techniques orthonormal vectors are derived from the high dimensional feature space, creating a new 'optimized' low-dimensional feature space. The authors mentioned that the goal of the ASO method is to detect the PC of the predictive structure. Knowing that PCA can be performed in two ways (t-mode and s-mode), the t-mode being the most frequently used by climatologist to identify recurrent spatial patterns over time, whereas the S-mode allows for identifying recurrent temporal patterns over space. How would the current method differ from an extended PCA in S-mode? I can imagine that using EPCA over a dataset as large as the one used here could be a real challenge for example. But I would like the authors to elaborate on the pros and cons of the new method as compared to already established techniques in the climate research such as PCA/EPCA for example.

**In the last paragraph of Sect. 2.4, we mentioned the main difference between the commonly-used PCA approaches and the proposed method. However, we have elaborated on the differences and potential advantages of our approach in the revised manuscript by extending Sect. 2.4 and adding relevant literature.**

p8.7: *The SVD-based ASO method can be interpreted as a dimensionality reduction technique applied to the model space (i.e., weights). It should be stressed here that this method must not be confused with principal component analysis (PCA), which is usually employed on the data space (input space of predictors) (Metzger et al., 2012; Ivits et al., 2014). The goal of the ASO method is to detect the principal components of the parameter matrix, while PCA identifies the principal components of the input data X. The goal of the ASO method can be achieved by considering the models of multiple tasks as samples of their own distribution. Therefore, these samples can only be formed by using an MTL approach, in which there is access to the models from multiple learning tasks. Moreover, in our work, we explicitly consider the climatic variables as predictors and the vegetation variable as target variable, and we learn the relationship between them in a supervised setting. As such, the regions that we define rely on the relationship between climate and vegetation in a prediction setting, and the clustering is calculated based on similarity of this relationship (i.e. the model coefficients for different locations), see Sect. 2.5 for more details. As such, we learn relationships between climate and vegetation in a supervised setting, whereas PCA-based methods (Metzger et al., 2012; Ivits et al., 2014) are fully unsupervised. In our study the SVD decomposition is used as part of the optimization algorithm, thus in a supervised setting. In this setting, the model weights are optimized based on a given training set. Therefore, the discovered structures are obtained during the training process.*

1.4) Sect. 2.5. The authors do not give any name or reference for the clustering technique used here. Please clarify if a new algorithm has been developed for the study or if an already developed clustering technique was applied.

**In the manuscript, it is mentioned that the clustering technique that we use is the agglomerative hierarchical clustering (with Euclidean distance measure). This is a well-known clustering method in Statistics (see Sect. 2.5 and 3.2 of the manuscript). To make it more clear to the broad audience of GMD, we have mentioned in the revised manuscript that we used the hierarchical clustering python implementation of scikit-learn, and added a specific reference.**

p10.32: *In our application, we use a hierarchical agglomerative clustering approach (Ward, 1963) where the number of clusters is not predefined.*

p17.26: *We use the implementation of Python for the L-BFGS optimizer, the singular value decomposition method and the hierarchical clustering (scikit-learn python library (Pedregosa et al., 2011)).*

1.5) General comment on the use of $R^2$ for assessing the model performance: at several occasions (in the manuscript and in the supplementary material), the authors used $R^2$ to quantify the performance of different models (MTL vs. STL, models with and without Granger causality, inclusion of higher-level features in the input dataset, final decision on the number of clusters). They generally conclude that the best model is the one with the highest $R^2$. I agree on the principle, however looking at the differences between $R^2$ (e.g. figures 3b and 3d, large areas present difference in $R^2$ below 0.1), I wonder whether all these differences are statistically significant. As based on the analysis of $R^2$, the authors are deciding on the final set of input data, the final design of the MTL model, and the final number of clusters, I would really urge the need for further statistical assessment of the model performances. One first analysis could simply be to estimate the percentage area of pixels with statistically significant increase in $R^2$.

**The distributions depicted in Figs. 3c and 3f of the manuscript show that the results of the MTL and STL methods are substantially different. Specifically, the distributions of the MTL results are shifted to the right, meaning that STL is outperformed by MTL at global scale. This result can be confirmed by any paired statistical test (Demšar, 2006). The results of the Wilcoxon non-parametric statistical test are included in the revised manuscript.**

**At pixel level, traditional statistical tests usually have too many assumptions for our purposes. Alternatively, non-parametric tests based on resampling, such as permutation or bootstrap tests, cannot really be applied due to the size of our data set. A proposed solution is to use the Diebold-Mariano statistical test (Diebold, 2015). This test has been used to compare the MTL and the STL approaches in the revised version of the manuscript.**

**For the final decision about the number of clusters, see our answer in comment 1.6.**

p13.6: *The dotted regions in Fig. 3b correspond to areas where the MTL model significantly outperforms the STL models based on the Diebold-Mariano statistical test, which compares model predictions (Diebold, 2015). For the statistical test, we use the False Discovery Rate (Benjamini and Hochberg, 1995) method to correct the p-values at level 0.05 due to the multiple-hypothesis testing setting.*

p13.16: *As it can be observed, the distribution of the $R^2$ scores is shifted to the right for the MTL, meaning that values are typically greater than those derived from the STL approach. Moreover, the skew towards the left in the blue histogram, with values close to zero, is an indication of the near-zero performance of the STL models in many locations. The Wilcoxon paired statistical test (Demšar, 2006) confirms that the results of the two approaches are statistically dif-*

*ferent (p-value $< 10^{-9}$).*

p13.27: *In this figure it becomes clear that climate dynamics Granger-cause monthly vegetation anomalies in most regions of the world, and the ability of the MTL model to detect deterministic relationships is evidenced. This is also confirmed by the Wilcoxon paired statistical test (p-value $< 10^{-9}$).*

p13.33: *Analogous to Fig. 3c, Fig. 3f compares the distributions of Granger causality (i.e., the difference in predictive performance in terms of $R^2$ between the full and the baseline model) between the STL and MTL approach. Once again, the blue histogram corresponds to the distribution of Granger causality retrieved using the STL approach, while the orange corresponds to the results of the MTL approach. The shift to the right of the orange histogram shows the larger ability of the MTL model to reveal Granger-causality between climate and vegetation. Similar to the previous comparison, the Wilcoxon paired statistical test (Demšar, 2006) confirms that the results of the two approaches are statistically different (p-value $< 10^{-9}$).*
*See Sect. 3.1 and Fig. 3b.*

Results

1.6) General comment on the final number of clusters: the fact that the majority of the Iberian Peninsula is included in the transitional energy driven cluster together with Ireland, an important part of SE Asia, part of Brasil and Venezuela Colombia makes me wonder if a higher number of clusters would not be more appropriate. The authors mentioned already in Figure S2 that the differences in the predictive performance for h = 6 - 15 are marginal. Further assessments should therefore be performed in order to identify the optimal number of hydro-climatic biomes. Part of this assessment should be dedicated to the understanding of the actual drivers (main predictors) for each biome. I believe providing a solid justification for the naming of the different biomes (by referring back to the main predictors) would be beneficial for the paper.

**We agree that the differences in predictive performance for h = 6-15 are marginal. However, the proposed method is a fully data-driven approach that is not fine-tuned based on any kind of prior knowledge. Therefore, the selection of the final value of the h parameter is based on an objective criterion, i.e. the model performance. As for the resulting map (Fig. 4a), although we are aware that this map may not fully reflect all particular expectations, we do believe that the spatial distribution broadly captures the expected regimes of climate–vegetation interactions, as described in the results section. Note as well that in our early experiments we ran our approach with a different number of clusters to visually inspect the resulting regions. The**

regions formed with h values close to 11 are similar to the reported ones (Fig. 4a of the manuscript). This result proves the robustness of the proposed method to detect the basic vegetation response types with respect to climate. The results (for h = 9-12) have been included as supplementary figures in the revised manuscript.

Concerning the label scheme, we should stress that the names of the biomes are inspired by the main predictors based on Papagiannopoulou et al. (2017). We are afraid that making the labels reflect these predictors more accurately would make them extremely complex and rather impractical. So we preferred to keep this label scheme in the revised manuscript.

p1.16 of Supplement: *Therefore, we can conclude that the method gives robust results as the strongest predictive structures are captured for the first most important components given by the singular value decomposition. This conclusion is also confirmed by Fig. S3, where the maps with 9 (Fig. S3a), 10 (Fig. S3b), 11 (Fig. S3c) and 12 (Fig. S3d) hydro-climatic biomes are depicted. In all figures, the tropics, the boreal and the arid regions are well-detected. In addition, sub-tropical regions and transitional ones are also commonly defined in all of the aforementioned figures. Differences in the borders of the identified regions are noticed between temperature-driven areas (e.g., Europe and North America). In transitional water- and energy-driven regions also there are some differences in the clusters borders. However, these inconsistencies can be explained by the smoother differences between the climatic and environmental conditions in these areas.*
*See Fig. S3 of Supplementary material.*

1.7) In relation to the previous comment, how does the new global map of hydro-climatic biomes perform as compared to previous ones (not including information of vegetation condition and response to climate)? It would be really interesting if the authors could showcase for one (or more) bio-climatic zone how the new bio-climatic zone provide a finer, more accurate picture of global terrestrial biomes by analysis the specific (/sub-local) ecosystem response to climate variability. To this regard, the bioclimatic map produced by Metzger et al. (see reference below) could also be of interest for comparison.
Metzger MJ, Bunce RGH, Jongman RHG et al. (2012) A high-resolution bioclimate map of the world: a unifying framework for global biodiversity research and monitoring. Global Ecology and Biogeography, 22, 630-638.

Thanks for the relevant reference, it has been cited in the revised manuscript. However, as we mentioned above, by using our approach we really aim for detecting regions of consistent behavior in response to climate (based on the learned weights). That is what we should evaluate. As such, we cannot really aim for 'accurate' biomes. This

is the reason why we do not compare our result to other data-driven approaches described in our introductory section that rely on climate and/or vegetation data (as Metzger et al. (2012)); our study tries to detect regions based on different criteria (based on the interaction between climate–vegetation and not on the data). This difference has been also stressed in the revised manuscript. We also note again that the comparison that is provided against traditional land classification schemes is to assure that comparable regions arise from our data-guided approach, despite these land cover types not being expecificaly prescribed. See also our answer to the remark 1.3 for the comparison of PCA-based methods to the proposed approach.

p2.11: *Metzger et al. (2012) applied an alternative data-driven approach on climate and vegetation data that used principal component analysis (PCA) to discover informative structures in the data. In this method, the principal components of the initial climate–vegetation data set were applied as input to a clustering algorithm.*

1.8) Figure 4. (c) The Köppen classification divides the world into 5 main classes and 29 sub-classes. The authors should justify the use of 10 classes in the figure. This can be very misleading when looking and interpreting the results. An example: I do not think that the statement p14, l21-23 '...the region of North Asia is coherent in terms of climate, but quite diverse in terms of vegetation types; the hydro-climatic biomes show a clear distinction from shrublands (...) to coniferous ...' holds entirely when looking at the high level details (29 classes) of the Köppen classification. Please justify your choice here.

It is true that the Köppen climate classification scheme consists of divisions and sub-divisions of the five main climate types. We could choose to use the divisions of the Köppen classification, which are basically 12 (if we also divide the tropics further) and not 10 as in Fig. 4. However, the use of 10 instead of 12 classes will not make the map look much different. Moreover, from the color scheme used in Fig. 4, it is clear that there are five main classes. In Fig. 4, we aim for comparing the regions detected by the proposed method to the regions based on the Köppen climate classification scheme. Since the division of 10 climate classes is closer to the number of regions detected by our approach, we choose this number of regions (10) on Köppen's map. Nonetheless, we agree that the statements mentioned in the comment sound a bit strong, so we modified them in the revised version. Again, the comparison to the Köppen and IGBP maps serves only as a general evaluation or proof of concept for our hydro-climatic biomes map, since in the end such maps are based on a different rationale. This has been clarified in the revised manuscript. We also visualized the distributions of the three classification schemes (Köppen, IGBP, hydro-climatic biomes) with respect to the mean annual precipitation and mean temperature. These scatter plots serve as a kind of "evaluation" for the statement that our hydro-climatic biomes map combines information from both the Köppen and IGBP maps.

p16.12: *For instance, the region of North Asia is quite coherent in terms of climate based on the 10 climate classes shown here (Fig. 4c), but quite diverse in terms of vegetation type (Fig. 4d); the hydro-climatic biomes show a clear distinction in the transition from shrublands (energy-driven) to coniferous forests (energy- and water-driven).*

p16.21: *The comparison to the Köppen-Geiger and IGBP maps serves only as a general evaluation or proof of concept for our hydro-climatic biomes map, since in the end such maps are based on a different rationale, and thus, there is no intent to 'outperform' these classification schemes. However, it can be observed in this comparison that the hydro-climatic biomes map in Fig. 4a combine information on climate and vegetation zones by illustrating regions where vegetation similarly interacts with the multi-month dynamics in climatic and environmental conditions. This conclusion is confirmed by the scatter plots in Figs. 4e-g. Figure 4e depicts our hydro-climatic biomes of Fig. 4a in climate space of mean annual temperature against precipitation, while Fig. 4f shows the same but for the Köppen-Geiger climate classes of Fig. 4c. In Fig. 4f, the five climate classes are well-separated, since their definition is based on these two climatic variables. On the other hand, Fig. 4g depicts the same information but for the IGBP map of Fig. 4d. In this figure, savannahs, tropics, and shrublands appear again well clustered. It can be observed that the scatter plot of Fig. 4e clearly lie between the two previous classifications in terms of clustering. Boreal biomes correspond to cold climate classes, the sub-tropical and mid-latitude water-driven biomes correspond to arid regions, while the transitional biomes correspond to the savannas and croplands.*

1.9) Supplementary material S4. The authors indicate that the best-formed clusters are depicted in FigS4a (hence by the hydro-climatic biomes). I find very difficult to make any final judgment of the best "depiction" (/detection) of biomes based on the 2-dimensional graphs provided here.

**We improved these figures in the revised manuscript by using the t-SNE method as dimensionality reduction technique.**

p5 of Supplement: *See Figs. S5 and S6 of the revised version in the Supplementary material.*

Technical comments

1.10) P5, l14: please add a reference for the statement: '...this kind of modelling is becoming more common in climate science...'

**This sentence refers to the previously mentioned studies, which are described in the same paragraph, and serves as a conclusion that MTL approaches are used more common recently than in the past in climate science. We added the relevant references in the sentence.**

p6.3: *Although this kind of modelling is becoming more common in climate science (i.e., Subbian and Banerjee (2013); McQuade and Monteleoni (2013); Gonalves et al. (2017); Xu et al. (2016)), it has not been combined (to the best of our knowledge) with clustering approaches in the context of mapping land cover nor climate-vegetation dynamics.*

1.11) P10, l10: please clarify what you mean by multi-month vegetation dynamics. Is it seasonal, subseasonal, yearly?

**We replaced "multi-month" with "mean monthly" vegetation dynamics.**

1.12) P12, l5: please correct 'Geanger' with 'Granger'

**Corrected.**

1.13) Figure 4. (a) the color code for the clusters sub-tropical energy driven and mid-latitude temperature driven are too similar. It is difficult to differentiate them. Please adjust the color scheme of the legend.

**Adjusted.**

1.14) p15, l22: The term 'turning point' has only been introduced recently in ecosystem and climate science so for clarity, you can refer to:
Horion S, Prishchepov AV, Verbesselt J, De Beurs K, Tagesson T, Fensholt R (2016) Revealing turning points in ecosystem functioning over the Northern Eurasian agricultural frontier. Glob Chang Biol, 22, 2801-2817.

**We included this relevant reference in the revised manuscript.**

**Response to the comments of Referee#2**

This study presents a new approach for the classification of global biomes. The idea is to focus on the statistical sensitivities of NDVI anomalies to multiple predictors. I do think that it is important to emphasize the "goal" of classification, and therefore the paper is a step in the right direction.

**We would like to thank the reviewer for the constructive feedback and thorough assessment. Below we provide a point-by-point response to each comment.**

2.1) I have, however, doubts if focusing on NDVI anomalies is the right target. In particular for tropical ecosystems NDVI does not tell us much about ecosystem dynamics and the figures show the underlying predictions are indeed not convincing. Hence, I have some doubts about the novelty that this classification can offer. Similar as all classical approaches, also this method fails to reveal the complex spatial patterns in tropical ecosystems. This is why I see this paper more as a methodological contribution that can actually help future studies to realize analogous exercises based on different data sets.

**We agree with the reviewer that although NDVI is a commonly-used index, it is known to saturate in tropical ecosystems. As we discussed in our previous work (Papagiannopoulou et al., 2017), the low predictive power of our model in tropical regions can be explained by the fact that in these regions, (i) the uncertainty in the data is larger, and (ii) vegetation might be primarily affected by other factors such as nutrient availability (rather than climate). However, with the proposed data-driven framework, pixels that belong to these tropical regions are grouped together. This means that the learned weight vectors of these pixels are similar and thus the clustering algorithm is able to detect these similarities to conform a coherent biome. Moreover, we also agree that our work can be seen as a methodological contribution, since it can be used in different application scenarios or with an alternative target variable. So, we are willing to explore the applicability of the method to a different target variable. As such, the applicability to microwave Vegetation Optical Depth (VOD) anomalies, instead of the NDVI anomalies, has been explored, see Fig. S7 in the Supplementary material. The result is quite close to the one in the manuscript (Fig.4a) (with the NDVI anomalies as target variable).**

p17.8:*Results for microwave vegetation optical depth (VOD) (Liu et al., 2011) anomalies as alternative to NDVI anomalies are consistent as shown in Supplementary material Fig. S7.*

2.2) Overall, the approach of the paper is to stack a series of methods. First, "Multi-Task Learning" is used to create a statistical prediction model whose sensitivities (condensed by SVD) later serve as basis for clustering. I applaud the authors for identifying a machine learning method that seems to capture spatial relationships. But my question is if there is no corresponding geostatistical approach out there that could be equally used (e.g. a GWR or so) which deals exactly with such questions? In particular, I believe (but don't know) that

the MTL does not consider the fact that lat-lon grid cells represent different geographical distances, or how do the authors considered that a global analysis is executed on a sphere?

**As we have described in the manuscript, our approach is purely data-driven. Therefore, we stress that we do not include any prior knowledge about the distances between the different pixels. On the contrary, we let the method learn the relationships between the different pixels. As such, the method may even group together remote pixels in which vegetation might have similar response to climate. Other geostatistical approaches, such as the GWR, assume that neighboring pixels have a similar behaviour with respect to specific variables. In these approaches, similarities between the pixels are learned by defining each time a single pixel as centroid and tuning the parameter of relatedness between this particular pixel and the surrounding pixels. In our work, we prefer to avoid this kind of neighborhood assumptions and focus on the discovery of relationships between the pixels based on the similarity in climate–vegetation interaction. We are also interested in methods that can be applied on large data sets with global coverage. However, we included this kind of methods as relevant work in the introduction of the revised manuscript.**

p2.23: *Previous studies rely on spectral information, supervised techniques or clustering approaches, which are applied to observations of climate variables and/or vegetation characteristics. However, these classification schemes are not based on the type of response of vegetation to climate dynamics. Recent advances in understanding vegetation response to climate variability highlight the importance of revealing the sensitivity of ecosystems to climate conditions, see Nemani et al. (2003); De Keersmaecker et al. (2015); Seddon et al. (2016); Papagiannopoulou et al. (2017b); Liu et al. (2018). Therefore, a step beyond these previous studies is a spatial characterization of the vegetation dynamics that are induced by climate variability, so that ecosystems of similar response to climate anomalies can be unveiled. This objective could be tackled by geostatistical approaches, such as geographically weighted regression (GWR) (Brunsdon et al., 1996), which assume that neighboring pixels have a similar behaviour with respect to specific variables; these methods have already been applied in studies with a regional focus (Propastin et al., 2008; Zhao et al., 2015; Georganos et al., 2017).*

2.3) The paper is neatly written, but I still had trouble finding my way through the paper. One aspect is that it is difficult to follow the paper without knowing the authors previous papers. In addition, I spent most of my time understanding Multi Task Learning. In particular section 2.4. was hard to understand. At this crucial point I would ask the authors to consider rewriting the paper in a

way that can be understood intuitively by environmental scientists who are not familiar with the method. Likewise the link to clustering is a bit opaque. What is a "hierarchical agglomerative clustering approach"? Etc.

**We expanded Section 2.4 to make the method more intuitive for the broad audience of GMD. Specifically, we provided additional explanations for the notation used in our model. For the clustering technique that we used, see our answer to remark 1.4.**

p9.9: *To clarify the notation used in the ASO method, we intuitively explain the symbolization of the method in relation to our specific setting; the problem of detecting locations with similar climate–vegetation dynamics. As mentioned above (Sect. 2.2 and 2.3), the input features that constitute the $\mathbf{X}^{(l)} \in \mathbb{R}^{N \times d}$ matrix consist of the climatic predictor variables, i.e., the extreme indices, lagged variables, etc., calculated based on raw climatic time series of a certain location $l$. The dimensions $N$ and $d$ correspond to the number of observations, i.e., the length of the time series and the number of predictor variables, respectively. The target variable for a particular location $l$, which is the NDVI anomalies, is symbolized with $\mathbf{y}^{(l)} \in \mathbb{R}^{N}$. As such, an observation of a certain location $l$ at a particular timestamp $i$ is denoted as a pair $(\mathbf{x}_i^{(l)}, y_i^{(l)})$. The goal of the ASO method is to learn the weight matrix $[\mathbf{w}^{(1)}, \mathbf{w}^{(2)}, ..., \mathbf{w}^{(L)}]$, i.e., a single weight vector $\mathbf{w}^{(l)}$ for each location $l$. This weight vector $\mathbf{w}^{(l)}$ is able to capture the relationship between the predictor variables and the target, i.e., the climatic variables and the NDVI anomalies. Therefore, climatic predictors that are more important for vegetation anomalies correspond to higher absolute values in the weight vector $\mathbf{w}^{(l)}$. As a result, locations with similar weights are considered as regions where vegetation responds to climate in a similar way. As described in a previous paragraph of this section, the ASO method assumes that the weight vectors $\mathbf{w}^{(l)}$ consist of two parts the $\mathbf{u}^{(l)}$ and the $\mathbf{v}^{(l)}\Theta$. These two parts are learned simultaneously in Algorithm 1 in an alternating fashion. The first part, i.e., the $\mathbf{u}^{(l)} \in \mathbb{R}^{d}$ belongs to the high-dimensional space, the initial one, which is equal to $d$. This part expresses the location-specific part of the weight vector, i.e., the deviation of each location's weight vector from the weights learned in a lower dimensional space. The second part consists of the matrix $\Theta \in \mathbb{R}^{h \times d}$ that represents the map from the initial dimension $d$ to the lower dimension $h$ and the weight vector $\mathbf{v}^{(l)} \in \mathbb{R}^{h}$. The map matrix $\Theta$ is common for all the locations (tasks) and can be learned across them due to the MTL approach. The weight vector $\mathbf{v}^{(l)}$ represents the projection of the initial weights to a low-dimensional space $h$. Intuitively, this second part of the weight decomposition expresses the coarsest and most important part of weights, since it detects the most important structures through the*

*map matrix $\mathbf{\Theta}$. The matrix $\mathbf{V} = [\mathbf{v}^{(1)}, ..., \mathbf{v}^{(L)}]^T \in \mathbb{R}^{L \times h}$ denotes the representation of the models in the low-dimensional space $h$ for the $L$ locations.*

2.4) What irritated me about the results is that the prediction method does not manage to explain more than 40% of the variance (why else would the scale in Fig. 3 a otherwise be cut off at $\geq 0.4$?). This is actually a bit disappointing and suggests that the regression model was not the right choice, or?

**In our study, the seasonal cycle from the NDVI time series is removed. Therefore, the task of predicting the NDVI anomalies is more difficult than just predicting the raw NDVI time series. This is due to the fact that the presence of autocorrelation in the NDVI anomalies time series is much lower. Note that if we target the raw NDVI time series (which includes the seasonal component), the $\mathbf{R}^2$ is close to 1 in most of the regions (Papagiannopoulou et al., 2017). In addition, it is worth noting that there are other factors – such as fires, harvesting, etc. – that affect vegetation dynamics but are not included in the data set. Therefore, we should be aware that we focus on explaining the variance of the NDVI anomalies, taking into account only climatic variables.**

Minor remarks:

2.5) The introduction does not provide a systematic overview of alternative approaches. Rather, we find here a rather random selection of climate and land cover classifications and the wording is not always correct. For example, the paper speaks of "big data" approaches, but I did not find any of the referenced studies really dealing with big data topics ("volume", "diversity", "speed", ...). I think we are talking here about (sometimes semi-heuristic), but essentially classical data exploration and machine learning methods. So, I think it would be nice to revise this part a bit to have a smooth start.

**In general, we would like to stress that the goal of our study is to provide a new data-driven methodology that can identify coherent regions in which vegetation responds to climate in a similar way. To the best of our knowledge, there are no other works that study this particular problem at global scale, with the arguable exception of the article pointed to by Referee#1 (Ivits et al., 2014). In addition, in the manuscript, we describe the most naive approach that one could follow by using single-task learning techniques (and by learning one model per pixel). In the Introduction, we provide an overview of the most related works to our study that indeed use machine learning methods and/or prior knowledge. We think that our work builds upon and goes one step further from previous efforts, such as the ones described in the Introduction, since it combines information**

from climate and vegetation and models the relationship between them. However, we added some relevant literature in the introductory section, see our answers to remarks 1.2 and 2.2. In addition, the term "big-data" has been replaced by "data-driven" in the revised manuscript.

p2.7: *As such, data-driven methods have become popular in their use for land cover and climate classifications.*

2.6) The paper is full of shortcuts such as "detrended seasonal NDVI anomalies", which are not as clear as they appear at first glance. I can think of a large number of possibilities for robustly estimating (linear/non-linear) trends and a further variety of methods for estimating seasonal cycles. It would be nice if such statements were more precise.

**We agree that these terms are not clearly described in the manuscript, and understand that the article should stand alone without the need of prior knowledge with regards to our previous work. We added additional statements to briefly describe this terminology in the revised manuscript.**

p3.28: *The target variable of our machine-learning framework is the de-trended seasonal NDVI anomalies. These are calculated through a simple linear de-trending and a multi-year average for each month of the year to capture the seasonal expectation  see Papagiannopoulou et al. (2017a) for more details. All other data sets, describing the multi-month local climate variability over the three-decade period, are used as predictor variables.*
*In addition, a wide range of 'high-level features' have been hand-crafted from the raw time series of predictors, and used as well as predictor variables. As such, our set of predictive features includes not just the raw data time series of each climate/environmental variable, but also: seasonal anomalies, de-trended seasonal anomalies, lagged variables, past cumulative variables, and extreme indices  see Papagiannopoulou et al. (2017a). The cumulative variables capture the climatic conditions up to present time; an example would be the precipitation of the last (e.g.) three months. Extreme indices include maximum/minimum values, consecutive dry days, values for specific percentiles, etc.*

2.7) The same comment applies to the selection of predictors e.g. seasonal anomalies, detrended seasonal anomalies, time delayed variables, and cumulative variables etc. look like a very arbitrary selection of predictors. In a paper that has a strong affinity to data-driven methods, I would expect a more formal variable selection following a clearly defined cost function. Maybe this is too late now, but still one question can be answered: why are these predictors all

regarded as "non-linear"? In most cases, they read like fairly linear transformations (maybe with the exception of cumulative variables).

**We refer the reviewer to our previous answer (2.6) for the first part of the comment. In addition, we would like to stress that our choice to use this set of predictors is based on the previous literature, as it has been analytically described in Papagiannopoulou et al. (2017). The constructed predictors are regarded as "non-linear", because their derivation from the raw data is not linear (see e.g. calculation of extreme indices). This has also been clarified in the revised manuscript.**

[revised manuscript text omitted]
. This conclusion is also confirmed by Fig. S3, where the maps with 9 (Fig. S3a), 10 (Fig. S3b), 11 (Fig. S3c) and 12 (Fig. S3d) hydro-climatic biomes are depicted. In all figures, the tropics, the boreal and the arid regions are well-detected. In addition, sub-tropical regions and transitional ones are also commonly defined in all of the aforementioned figures. Differences in the borders of the identified regions are noticed between temperature-driven areas (e.g., Europe and North America). In transitional

[Figure]

**(a)** Explained variance (R²) of the MTL model (raw predictors)

**(b)** Difference (R²)

**Figure S1.** Comparison of the predictive performance in terms of $R^2$ of the model which does not include the cumulative variables and the extreme indices with the model which is trained with the full collection of higher-level features (Papagiannopoulou et al., 2017a). (a) Explained variance ($R^2$) of NDVI anomalies based on the raw data of the climatic variables as well as their 6-lagged values (cumulative variables and the extreme indices are not included as predictors to the model). (b) Difference in terms of $R^2$ between the model without cumulative and extreme predictors and the model which includes all the higher-level feature representation in Fig. 3a of the manuscript.

[Figure]

**Figure S2.** Assessing the number of biomes: Median of the predictive performance of the ASO-MTL model in terms of $R^2$ when the value of the $h$ parameter varies. For $h = 11$ the model scores the maximum value of $R^2$. However, the differences in the predictive performance for $h = 6, [...], 15$ are marginal.

water- and energy-driven regions also there are some differences in the clusters' borders. However, these inconsistencies can be explained by the smoother differences between the climatic and environmental conditions in these areas.

[Figure]

**Figure S3.** Maps with different number of hydro-climatic biomes. (a) $h = 9$ (i.e., 9 hydro-climatic biomes) , (b) $h = 10$, (c) $h = 11$ (Fig. 4a of the manuscript), and (d) $h = 12$.

**S3 Visualization of the most important predictive structures**

In Sect. 2.5 of the manuscript, we describe the steps of the SVD-based ASO algorithm, which learns a low-dimensional feature representation for our tasks based on the relationships between them. The learned matrix $\boldsymbol{\Theta}$ maps the high-dimensional space to a (lower) $h$-dimensional space, storing the loads of the original weights to the "highly predictive structures". Thus, the task models are also projected to this shared lower-dimensional space. This information is stored in the matrix $\mathbf{V}$ on which the clustering approach is performed. Figure S4 presents the values of the tasks in the first 6 components of the matrix $\mathbf{V}$. Similar pixel values to the same components mean similar  climate–vegetation dynamics. There are several remarks considering Fig. S4: (1) all the 6 components are able to distinguish specific regions according to different criteria such as regions with temperate and dry climate, regions with cold and dry climate, tropical and dry climate, etc.; (2) pixels which are grouped into the same region in the final clustering result (Fig. 4a of the manuscript) tend to have similar values in a particular

[Figure]

**Figure S4.** Visualization of the first 6 "principal components" of the predictive structures. The classification of the land surface into the hydro-climatic biomes is based on the importance of these structures for each location. The color intensity in the map indicates the value magnitude of each pixel in a particular predictive structure.

predictive structure, and (3) the differences in the values across regions are intense, and in some cases one can recognize the boundaries of the regions depicted in Fig. 4a of the manuscript.

**S4   Visualization of the predictive structures with the different land surface classifications**

As in Zscheischler et al. (2012), we conduct a dimensionality reduction to the matrix $V$ which contains the clustering data. We
5   separately present the results for the Northern and the Southern Hemisphere (ibid.) – see Figs. S5 and S6, respectively. The data is projected onto the first 2  components of the t-SNE method (Maaten and Hinton, 2008) and visualized based on the  hydro-climatic biomes (Fig.S5a and S6a), the  Köppen-Geiger clustering (Köppen, 1936) (Fig.S5b and
10   S6b) and the  IGBP clustering (Loveland and Belward, 1997) (Fig.S5c and S6c). We use the same color representation as in Fig. 4a of the manuscript. That way we can assess if the learned predictive structures match well the classes of the different classification schemes.

Considering Fig. S5, one can see that the best-formed clusters are depicted in Fig. S5a, as the clustering has been performed on this dataset (as expected). Figure S5 c represents the IGBP land use classification; the
15   tropical regions are well-detected as well as the forest- and the cropland-covered regions. This means that the learned predictive

[Figure]

[Figure]

**Figure S5.** Data projection to the first  two t-SNE components for the Northern Hemisphere. Each point represents one pixel of the global grid and it is colored based on (a) the hydro-climatic biomes, (b) the  Köppen-Geiger climate classification, and (c) the IGBP land use classification. For the color-class mapping see Fig. 4 of the manuscript.

structures are highly relevant to the vegetation type of each region. In addition, Fig. S5 b indicates that the cold, the arid and the tropical regions can be well distinguished by the learned structures whereas the temperate climate is scattered among the others and is thus harder to  be identified.

Figure S6 depicts the same plots for the Southern Hemisphere. As in Zscheischler et al. (2012), overall, the classes identified by the various classification schemes show a worse match than for the Northern Hemisphere. However, Fig. S6a shows that the predictive structures can clearly distinguish the sub-tropical water-driven region and the transitional energy/water-driven regions as well. In addition, the Köppen-Geiger climate classes (Fig. S6b) of the tropic and the arid regions are also identified in a certain degree. The IGBP classes, in Fig S6c, do not form clear clusters.

[Figure]

[Figure]

**Figure S6.** As Fig. S5 but for the Southern Hemisphere.

[Figure]

**Figure S7.** Hydro-climatic biomes based on vegetation optical depth (VOD) data. The VOD anomalies used as target variable in the proposed approach.

Papagiannopoulou, C., Miralles, D. G., Decubber, S., Demuzere, M., Verhoest, N. E. C., Dorigo, W. A., and Waegeman, W.: A non-linear Granger-causality framework to investigate climate–vegetation dynamics, Geosci. Model Dev., 10, 1945–1960, https://doi.org/10.5194/gmd-10-1945-2017, 2017a.

5 Papagiannopoulou, C., Miralles, D. G., Dorigo, W. A., Verhoest, N. E. C., Depoorter, M., and Waegeman, W.: Vegetation anomalies caused by antecedent precipitation in most of the world, Environ. Res. Lett., 12, 074 016, https://doi.org/10.1088/1748-9326/aa7145, 2017b.

Zscheischler, J., Mahecha, M. D., and Harmeling, S.: Climate classifications: the value of unsupervised clustering, Procedia Comput. Sci., 9, 897–906, 2012.